# *Proline rich 11* (*PRR11*) overexpression amplifies PI3K signaling and promotes antiestrogen resistance in breast cancer

Kyung-min Lee [1], Angel L. Guerrero-Zotano [2], Alberto Servetto[1], Dhivya R. Sudhan[1], Chang-Ching Lin[1], Luigi Formisano[2], Valerie M. Jansen[2], Paula González-Ericsson [3], Melinda E. Sanders[3], Thomas P. Stricker[3], Ganesh Raj[4], Kevin M. Dean [5], Reto Fiolka [5,6], Lewis C. Cantley [7], Ariella B. Hanker [1] & Carlos L. Arteaga [1,3]✉

The 17q23 amplicon is associated with poor outcome in ER[+] breast cancers, but the causal genes to endocrine resistance in this amplicon are unclear. Here, we interrogate transcriptome data from primary breast tumors and find that among genes in 17q23, *PRR11* is a key gene associated with a poor response to therapeutic estrogen suppression. PRR11 promotes estrogen-independent proliferation and confers endocrine resistance in ER[+] breast cancers. Mechanistically, the proline-rich motif-mediated interaction of PRR11 with the p85α regulatory subunit of PI3K suppresses p85 homodimerization, thus enhancing insulin-stimulated binding of p110-p85α heterodimers to IRS1 and activation of PI3K. *PRR11*-amplified breast cancer cells rely on *PIK3CA* and are highly sensitive to PI3K inhibitors, suggesting that *PRR11* amplification confers PI3K dependence. Finally, genetic and pharmacological inhibition of PI3K suppresses PRR11-mediated, estrogen-independent growth. These data suggest ER[+]/*PRR11*-amplified breast cancers as a novel subgroup of tumors that may benefit from treatment with PI3K inhibitors and antiestrogens.

---

[1] Harold C. Simmons Comprehensive Cancer Center, University of Texas Southwestern Medical Center, Dallas, TX 75390, USA. [2] Department of Medicine, Vanderbilt University Medical Center, Nashville, TN 37232, USA. [3] Breast Cancer Research Program, Vanderbilt Ingram Cancer Center, Vanderbilt University Medical Center, Nashville, TN 37232, USA. [4] Department of Urology, University of Texas Southwestern Medical Center, Dallas, TX 75390, USA. [5] Department of Cell Biology, University of Texas Southwestern Medical Center, Dallas, TX 75390, USA. [6] Lyda Hill Department of Bioinformatics, University of Texas Southwestern Medical Center, Dallas, TX 75390, USA. [7] Meyer Cancer Center, Weill Cornell Medicine College, New York, NY 10065, USA. ✉email: Carlos.Arteaga@UTSouthwestern.edu

Approximately 80% of breast cancers are estrogen receptor (ER)-positive and depend on estrogen for growth[1]. Therapies for ER+ breast cancer inhibit ER signaling by directly antagonizing ER (i.e., fulvestrant) or by abolishing estrogen production (i.e., aromatase inhibitors). Adjuvant anti-ER therapies significantly reduce the risk of recurrence in patients with ER+ breast cancer[2]. However, approximately 20% of patients treated with adjuvant endocrine therapy eventually relapse with metastatic disease[3]. To date, several mechanisms of de novo and acquired resistance to endocrine therapy have been reported[4]. Due to advances in large-scale tumor DNA sequencing, several somatic alterations that promote endocrine resistance have been discovered. Mutations in the ligand-binding domain of ESR1, the gene encoding ERα, confer resistance to estrogen suppression via ligand-independent ERα transcriptional activity[5]. Amplification of growth factor receptors such as ERBB2 and FGFR1 has also been associated with endocrine therapy resistance[6,7]. Enrichment of CCND1 amplification in luminal B tumors also suggests a potential causal role with a drug-resistant phenotype[1]. More recently, Razavi and colleagues reported that mutations in components of the mitogen-activated protein kinase (MAPK) pathway and the ER transcriptional program, found in approximately 20% of ER+ breast cancers, are associated with shorter response to antiestrogen therapy[8]. Preclinical and clinical studies have suggested a critical role for hyperactivation of the phosphoinositide 3-kinase (PI3K)/AKT pathway in endocrine resistance[9–12]. In line with this causal role, the PI3Kα inhibitor alpelisib in combination with the ER antagonist fulvestrant was clearly superior than fulvestrant alone in patients with advanced ER+/PIK3CA mutant breast cancer[13], leading to the approval of alpelisib + fulvestrant in this subgroup of ER+ breast cancers.

We recently reported genomic profiling of ER+ breast tumors after short-term treatment with the aromatase inhibitor (AI), letrozole[14]. In this study, the 11q13.3, 8p11.23, and 17q21-23 amplicons significantly correlated with high levels of the proliferation marker Ki67 upon drug-induced estrogen suppression. FGFR1 and CCND1 amplification, in 8p11-12 and 11q13, respectively, were associated with resistance to letrozole as defined by maintenance of a high Ki67 score on treatment. Although the 17q23 amplicon has been associated with highly proliferative luminal B tumors and high risk of recurrence in ER+ breast cancers[15,16], a specific gene or genes in this region that would be causal to endocrine resistance have not been uncovered. In a recent study, we performed whole transcriptome analysis on RNA extracted from 58 ER+ breast cancers from patients treated with prolonged neoadjuvant letrozole[17]. In this cohort, we identified PRR11 (Proline rich 11), a protein-coding gene located in chromosome 17q22-23, to be overexpressed in tumors resistant to estrogen suppression compared to letrozole-sensitive tumors. PRR11 has been implicated in poor outcome of various cancer types[18–20], but the molecular basis for this association is unclear. We hypothesized that PRR11 amplification in the 17q23 amplicon promotes endocrine resistance in ER+ breast cancer. We show herein that high PRR11 is causally associated with estrogen-independent growth of ER+ breast cancer cells. This action involved a PR (proline rich) domain-dependent interaction of PRR11 with the p85 regulatory subunit of PI3K which reduces homodimerization of p85 and, in turn, is permissive of ligand-induced association of p110α with insulin receptor substrate 1 (IRS1) and activation of PI3K. Ectopic expression of PRR11 failed to promote estrogen-independent growth when p110α was knocked down, and PRR11-overexpressing cells were highly sensitive to PI3K inhibitors, suggesting that PRR11 amplification generates dependence on PI3K signaling, particularly in the setting of estrogen deprivation. Taken together, these data suggest a combination of PI3K and ERα targeted therapies is

a rational approach against ER+ breast cancers with PRR11 amplification.

## Results

**PRR11 is associated with poor outcome of ER+ breast cancers.** In order to identify genes associated with poor outcome of ER+ berast cancers treated with antiestrogens, we had performed whole transcriptome analysis on RNA extracted from 58 ER+ breast cancers from patients treated with long-term letrozole for a median of 7.2 months (Supplementary Table 1; cohort of Guerrero-Zotano et al.[17]). PRR11 mRNA was significantly upregulated in resistant tumors. In this study, resistance to estrogen suppression was defined by a preoperative endocrine prognostic index (PEPI) ≥ 4 and/or evidence of cancer relapse after a median follow-up of 5 years [log2 fold change > 1 and false discovery rates (FDR) < 0.05; Fig. 1a]. RNA-seq analysis of the treated tumors showed that among 51 genes in 17q23, PRR11 was the only gene significantly overexpressed in resistant vs. sensitive cancers (Log$_2$ fold change = 1.15, FDR = 0.004, $p$ = 1.02E-05; Fig. 1b). Patients with PRR11-high cancers displayed an increased risk of relapse in the same cohort [hazard ratio (HR) = 3.753; 95% confidence interval (CI), 1.045–13.47; Fig. 1c]. In the Kaplan–Meier Plotter database[21], high PRR11 mRNA levels were also associated with a shorter relapse-free survival (RFS) of ER+/HER2− breast cancers treated with endocrine therapy (HR = 3.85; 95% CI, 1.95–7.59; Fig. 1d), but this association was not present in patients with HER2+ or triple-negative breast cancer (TNBC; Supplementary Fig. 1a). We further interrogated the association of PRR11 expression with response to estrogen suppression in two other clinical studies of ER+ breast cancers treated with a neoadjuvant aromatase inhibitor (Supplementary Table 1; cohort of Giltnane et al.[14]; cohort of Miller et al.[22]). In these studies, maintenance of a high Ki67 index on treatment was used as a surrogate of resistance to estrogen suppression. In these two cohorts, we also found a statistically significant correlation between PRR11 mRNA and on-treatment high Ki67 levels (Fig. 1e, f). To correlate PRR11 expression with response to estrogen suppression, PRR11 protein levels were evaluated by IHC in tumor sections from a previously reported cohort of 175 ER+ breast cancers treated with letrozole before surgery (NCT00651976[14]). Based on the exponential curve of PRR11 protein levels, tumors were classified as PRR11 ≤ 1% (negative), 1–15% (positive) and >15% (high; Supplementary Fig. 1b). PRR11 protein levels were statistically higher in tumors with poor response to letrozole, as defined by on-treatment high Ki67 levels[14] (Fig. 1g, h).

PRR11 has been reported as an estrogen-responsive gene[23]. Thus, we examined the possibility that high PRR11 mRNA remained high in resistant tumors because ER is not sufficiently suppressed by letrozole treatment. In another cohort of 18 ER+ breast tumors treated with an aromatase inhibitor (cohort of Miller et al.[22]), PRR11 mRNA was not downregulated in post-treatment compared to pre-treatment tumors (Supplementary Fig. 1c), implying PRR11 is not regulated by ligand-induced ER in vivo. Moreover, PRR11 mRNA levels did not correlate with estrogen response gene set signatures in three cohorts of tumors treated with an aromatase inhibitor (cohorts of Guerrero-Zotano et al., Giltnane et al., and Miller et al.; Supplementary Fig. 1d). Consistent with these correlations, exogenous estrogen did not increase PRR11 mRNA levels in MCF7 and HCC1428 cells (Supplementary Fig. 1e).

**PRR11 is a key gene associated with endocrine resistance.** The cytogenetic band of PRR11 is designated 17q22 or 17q23.2 (Ensembl or HGNC, respectively), located at the terminal region

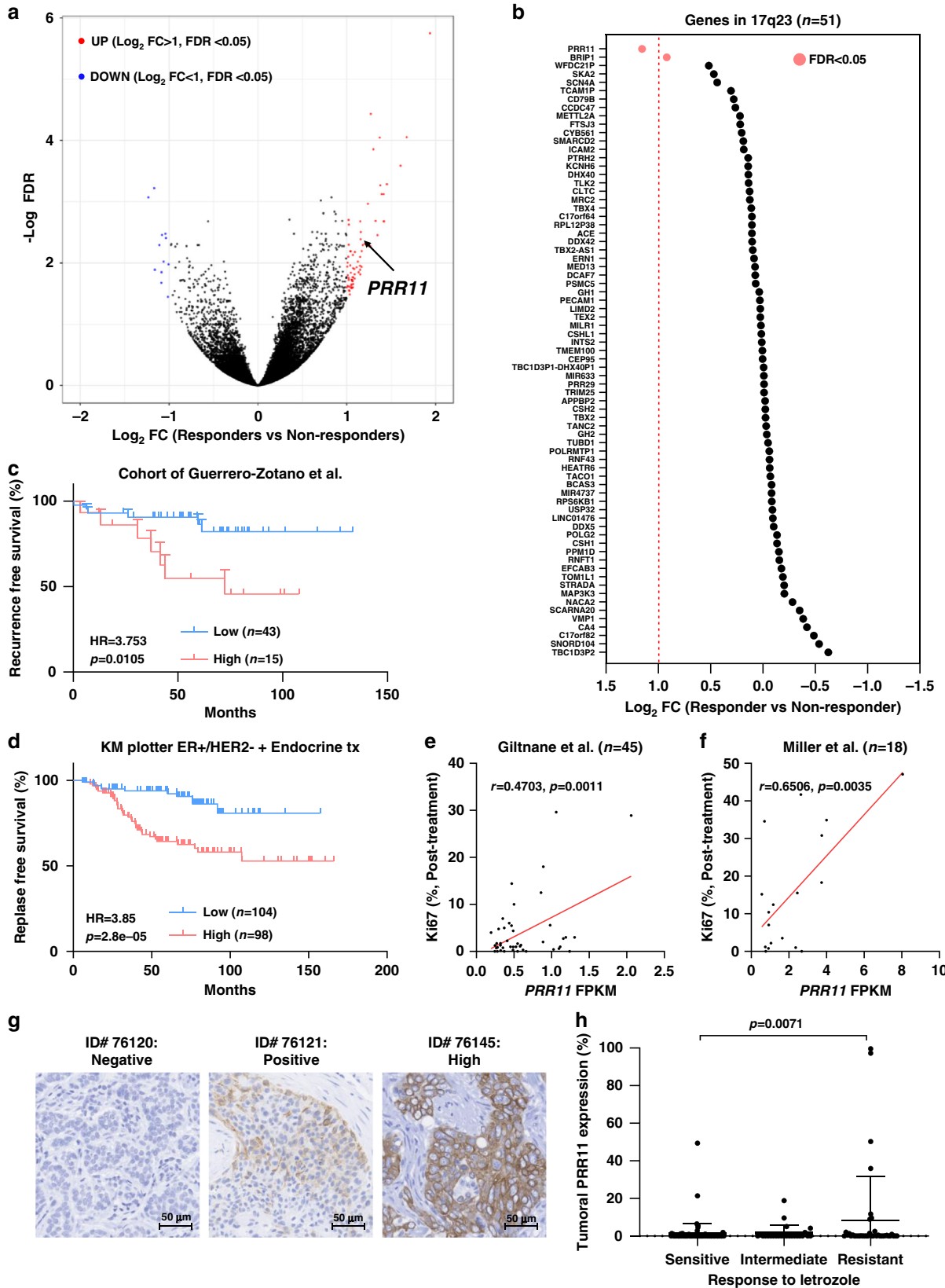

of 17q22 close to the 17q23 region (Supplementary Fig. 2a). The rate of *PRR11* amplification is 15.9% in the Metastatic Breast Cancer (MBC) project, but 9.5% and 9.4% in METABRIC and The Cancer Genome Atlas (TCGA), respectively (Fig. 2a). These

last two cohorts comprise primary breast tumors, thus suggesting a higher rate of *PRR11* amplification in metastatic (in the MBC project) compared to ER$^+$ primary breast tumors. Of note, most metastatic ER$^+$ breast cancers have undergone adjuvant

**Fig. 1 PRR11 is associated with poor clinical outcome of ER$^+$ breast cancers treated with antiestrogens. a** Volcano plot of genes differentially expressed in non-responding tumors compared to responding tumors. Log$_2$ fold change (FC) and false discovery rates (FDR) were calculated using DeSeq2 package. **b** Log$_2$ FC of 17q23 locus genes in ER$^+$/HER2$^-$ breast cancers treated with long-term letrozole ($n = 51$). Genes in 17q23 locus were selected based on the Atlas of Genetics and Cytogenetics[25]. Red bars indicate genes with FDR < 0.05. **c** Recurrence-free survival in ER$^+$ breast cancers, treated with long-term neoadjuvant letrozole, with low or high *PRR11* mRNA levels. A *PRR11* FPKM cut-off (3.93) obtained from the human protein atlas was used to divide *PRR11*-high ($n = 43$) and -low ($n = 15$) tumors (https://www.proteinatlas.org/ENSG00000068489-PRR11/pathology/breast+cancer). The Mantel–Cox model was used to calculate the hazard ratio (HR) and p value. **d** Relapse-free survival of ER$^+$/HER2$^-$ breast cancers, treated with endocrine therapy, with low ($n = 104$) or high ($n = 98$) *PRR11* mRNA levels by the auto select best cutoff in Kaplan–Meier Plotter. HR and p were adopted from the Kaplan–Meier Plotter (http://kmplot.com/analysis/). **e, f** Correlation between the on-treatment percent of Ki67$^+$ tumor cells and *PRR11* mRNA level in breast tumors from the cohort of Giltnane et al. and Miller et al. (Pearson correlation). **g** Representative PRR11 immunohistochemistry (IHC) images of primary ER$^+$ breast tumors. **h** PRR11 levels were plotted as a function of response to estrogen suppression with letrozole in trial NCT00651976[14]. Data represent the mean ± SD ($n = 91, 25, 39$ for drug sensitive, intermediate and resistant group, respectively; two-tailed unpaired t-tests). Source data are provided as a Source data file.

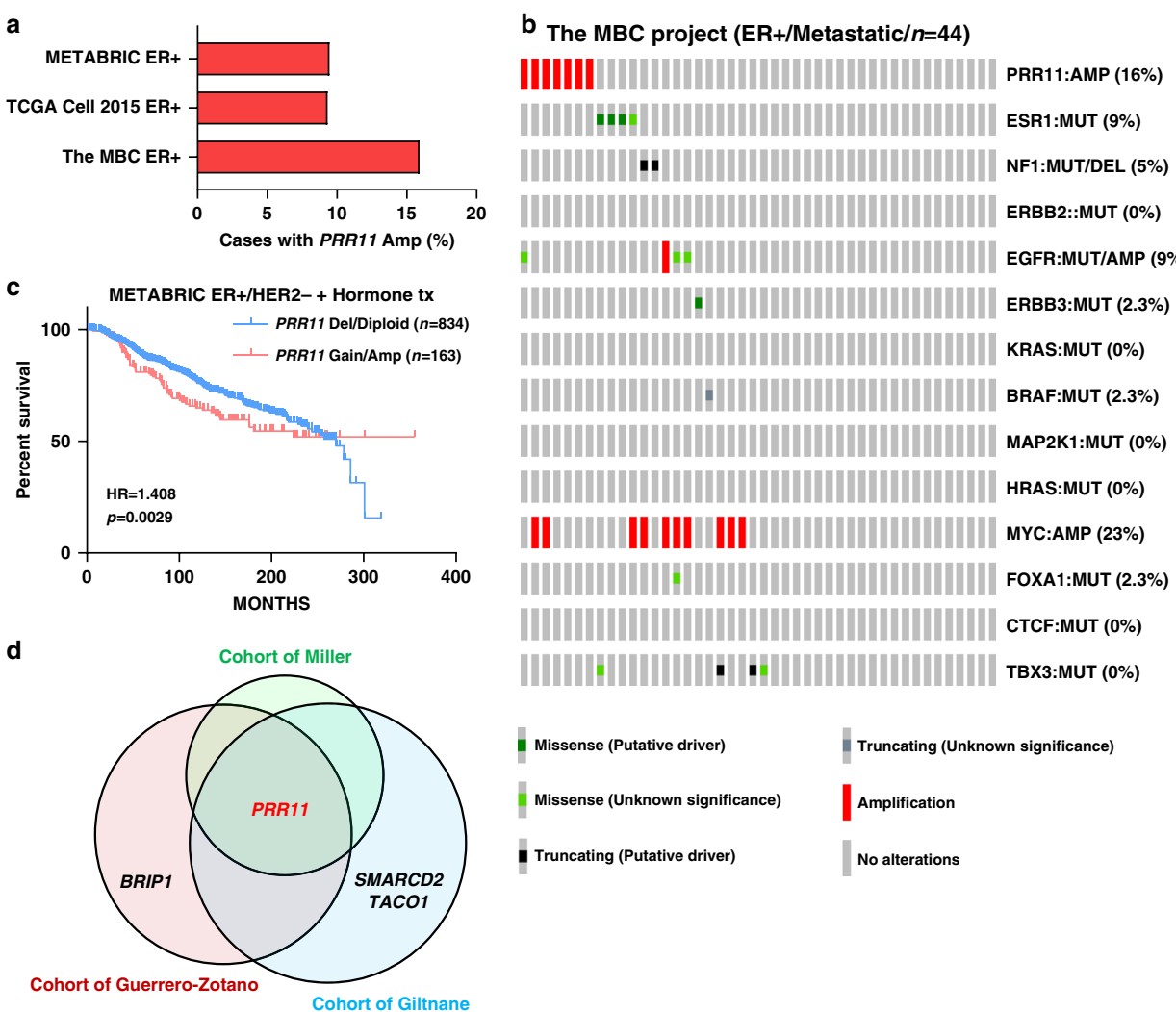

**Fig. 2 PRR11 is a key gene in 17q23 associated with endocrine resistance. a** Frequency of *PRR11* amplification in MBC project, TCGA or METABRIC ER$^+$ breast cancers. **b** Oncoprint of putative endocrine-resistant drivers in metastatic ER$^+$ tumors from MBC project dataset. ER$^+$ tumors classified as 'METASTATIC DISEASE PRESENT' were interrogated ($n = 44$). **c** Plot of disease-free survival of METABRIC ER$^+$/HER2$^-$ breast cancer patients treated with anti-hormone therapy as a function of *PRR11* copy number [gain/amplification ($n = 163$) vs. diploid/deletion ($n = 834$)]. **d** Venn diagram of genes in 17q23 that significantly correlated with on-treatment Ki67 levels (Pearson $r > 0.4$; $p < 0.05$). Source data are provided as a Source data file.

treatment with endocrine therapy, so metastatic ER$^+$ tumors often harbor somatic alterations associated with endocrine resistance. In ER$^+$ tumors in the MBC project dataset, *PRR11* amplification was mutually exclusive with mutations in *ESR1* and *NF1*, both established mechanisms of antiestrogen resistance (Fig. 2b). In the METABRIC cohort, *PRR11* copy number gain or

amplification predicted shorter disease-free survival of patients with ER$^+$/HER2$^-$ breast cancer treated with antiestrogens (HR = 1.408; 95% CI, 1.029–1.926; Fig. 2c). Also in METABRIC, *PRR11* copy number alterations (CNAs) were correlated with *PRR11* mRNA expression (Supplementary Fig. 2b). In a panel of 56 breast cancer cell lines in the CCLE dataset[24], high *PRR11*

 

copy number was significantly correlated with *PRR11* mRNA (Supplementary Fig. 2c). Moreover, we observed that *PRR11*-amplified breast cancer cell lines express higher levels of PRR11 protein compared to non-amplified cell lines ($n = 11$; Supplementary Fig. 2d). We finally verified *PRR11* amplification by fluorescence in situ hybridization (FISH) in a letrozole-resistant primary breast cancer (Supplementary Fig. 2e).

The gene(s) in the 17q23 amplicon that are causal to endocrine resistance have not been identified. Using the Atlas of Genetics and Cytogenetics[25], we found 90 genes located in the 17q23 amplicon (Supplementary Table 2). We next examined the association of these genes with on-treatment Ki67 levels in three clinical studies of ER$^+$ breast cancers treated with a neoadjuvant aromatase inhibitor (Supplementary Table 1; cohort of Guerrero-Zotano[17]; cohort of Giltnane[14]; cohort of Miller[22]). This analysis revealed that four of these 90 genes in 17q23 (*PRR11, BRIP1, SMARCD2,* and *TACO1*) correlated statistically with on-treatment Ki67 levels (Pearson $r > 0.4$, $p < 0.05$; Fig. 2d; Supplementary Tables 3–5). Across these three studies, *PRR11* was the only one of these four genes that exhibited a significant correlation with a high Ki67 score. Finally, among 67 genes in 17q23, high *PRR11* mRNA levels also correlated with a shorter RFS in patients with ER$^+$/HER2$^-$ breast cancer treated with endocrine therapy in the Kaplan–Meier Plotter database (Supplementary Fig. 2f). Collectively, these data suggest that *PRR11* may be a key gene in 17q23 associated with endocrine resistance.

Several genes in the 17q23 amplicon have been speculated to be associated with poor outcome in breast cancer[15,26]. To determine whether *PRR11* is an essential gene in 17q23-amplified breast cancer cells, we interrogated genome-scale RNAi screening data of MCF7 cells that harbor 17q23 amplification[27] in Project Achilles dataset (v2.4.3)[28]. shRNAs targeting 47 genes in 17q23 were screened in Project Achilles dataset. Of these 47 genes, *PRR11* displayed the lowest Analytic Technique for Assessment of RNAi by Similarity (ATARiS)[29] score, thus implying a high dependency of 17q23-amplified breast cancer cells on *PRR11* (Supplementary Fig. 2g).

**PRR11 overexpression confers resistance to antiestrogens**. To further prioritize genes in 17q23 associated with resistance to estrogen suppression, we transduced *PRR11, BRIP1, SMARCD2,* and *TACO1* into MDA-MB-134VI and MDA-MB-175VII cells, which do not harbor 17q23 amplification (Fig. 3a). *PRR11* was the only gene that promoted growth of both cell lines under conditions of estrogen deprivation (Fig. 3b). Next, we employed MCF7 LTED (long-term estrogen deprived) and HCC1428 LTED cells[9]. MCF7 and HCC1428 wild type cells harbor *PRR11* high copy number[24]. *PRR11* ablation by siRNA abolished estrogen-independent growth of MCF7 LTED and HCC1428 LTED cells (Fig. 3c). The inhibitory effect of *PRR11* ablation was rescued by re-expression of *PRR11* (Fig. 3d, e). To assess the role of PRR11 in vivo, we generated xenografts of MCF7 cells stably expressing a doxycycline-inducible *PRR11* shRNA in ovariectomized athymic mice. Treatment with doxycycline markedly reduced PRR11 protein levels and arrested growth of established MCF7 xenografts expressing *PRR11* shRNA but not a control shRNA (Fig. 3f and Supplementary Fig. 3a).

In the PRISM repurposing 19Q3 dataset[30], *PRR11*-amplified ER$^+$ breast cancer cell lines displayed a lower sensitivity to the ER antagonist fulvestrant compared to cells without *PRR11* amplification (Fig. 3g). Transduction of *PRR11* into MDA-MB-134VI and MDA-MB175VII cells attenuated growth inhibition by fulvestrant (Fig. 3h and Supplementary Fig. 3b). *PRR11* knockdown re-sensitized fulvestrant-resistant (FulvR) MCF7 cells to fulvestrant (Fig. 3i and Supplementary Fig. 3c). A similar result

was observed in tamoxifen-resistant (TamR) MCF7 and TamR HCC1428 cells (Supplementary Fig. 3d). These data suggest PRR11 promotes resistance to antiestrogens and its down-regulation enhances the action of ER-targeted therapies against ER$^+$ breast cancer cells.

**PRR11 promotes proliferation of ER$^+$ breast cancer cells**. We next analyzed RNA-seq data in the cohort of Guerrero-Zotano et al.[17] using 125 previously reported breast cancer-related gene expression signatures[31]. Nine proliferation-associated signatures were significantly enriched in tumors with high *PRR11* mRNA expression (FDR < 0.01; Fig. 4a). Hallmark gene sets associated with proliferation, including "E2F_TARGETS" and "G2M_CHECKPOINT" were significantly enriched in ER$^+$/HER2$^-$ tumors with *PRR11* gain or amplification in METABRIC and in ER$^+$ tumors with *PRR11* amplification in TCGA (Supplementary Fig. 4a, b).

To inquire further into the mode of action of PRR11 in cell proliferation, we determined the expression of 84 cell cycle-associated genes using a PCR array. Six genes were reduced upon transfection of *PRR11* siRNA into MCF7 LTED cells (*SKP2, CDKN1A, CCNB2, CCNA2, CKS2,* and *CCNB1*; FC < 0.5; Fig. 4b). With the exception of *SKP2* and *CDKN1A*, expression of each of these genes was significantly elevated in ER$^+$ breast cancers with *PRR11* copy number gain/amplification in the METABRIC and TCGA dataset (Supplementary Fig. 4c, d). *PRR11* ablation arrested cell cycle progression and inhibited proliferation in MCF7 LTED, MCF7 TamR, MCF7 FulvR, and HCC1428 LTED cells (Fig. 4c and Supplementary Fig. 4e) and these were rescued by re-expression of *PRR11* (Fig. 4d). Conversely, *PRR11* over-expression resulted in a marked increase in cells in S phase in estrogen-deprived MDA-MB-134VI and MDA-MB-175VIII cells (Fig. 4e). Consistent with a cytostatic effect, *PRR11* knockdown reduced RB phosphorylation levels (Fig. 4f). Even in absence of estrogen, ectopic expression of *PRR11* was capable to induce RB phosphorylation (Fig. 4g). Finally, ER transcriptional activity measured with an estrogen response element (ERE)-luciferase reporter was not affected by siRNA-mediated *PRR11* ablation in parental and LTED MCF7 and HCC1428 cells (Supplementary Fig. 4f), suggesting that PRR11 confers resistance to estrogen suppression via growth-promoting signaling pathways independent of ER. PRR11 ablation did not suppress proliferation in PRR11-overexpressing triple negative and HER2$^+$ breast cancer cells (Supplementary Fig. 4g, h).

**PRR11 reduces p85 homodimers and enhances PI3K activation**. Gene set signature analysis shown in Fig. 4a also revealed that PIK3CA and IGF1 signaling pathways, both reported to be associated with endocrine resistance[9,32], were significantly enriched in *PRR11*-high tumors. In line with this association, *PRR11* knockdown resulted in a reduction in phosphorylated AKT (p-AKT) and p110α protein in LTED, TamR, and FulvR cells (Fig. 5a). AKT inactivates GSK3β through the phosphorylation, which in turn, promotes the stabilization of cyclin D1 protein[33,34]. Indeed, *PRR11* knockdown resulted in an inactivation of GSK3β (Fig. 5a). *PIK3CA* mRNA levels were not affected by *PRR11* ablation, suggesting that *PIK3CA* transcription is not regulated by PRR11 (Supplementary Fig. 5a). A similar reduction in p-AKT by *PRR11* ablation were observed in MCF7 cells whose *PIK3CA* E545K allele had been corrected to a wild type sequence by somatic cell gene targeting, implying that the effect of PRR11 in PI3K activity is not limited to cells with *PIK3CA* mutations (Supplementary Fig. 5b–d). Conversely, overexpression of *PRR11* resulted in an increase of p-AKT and p-GSK3β levels in MDA-MB-175VII and MDA-MB-134VI cells (Fig. 5b). To determine

 

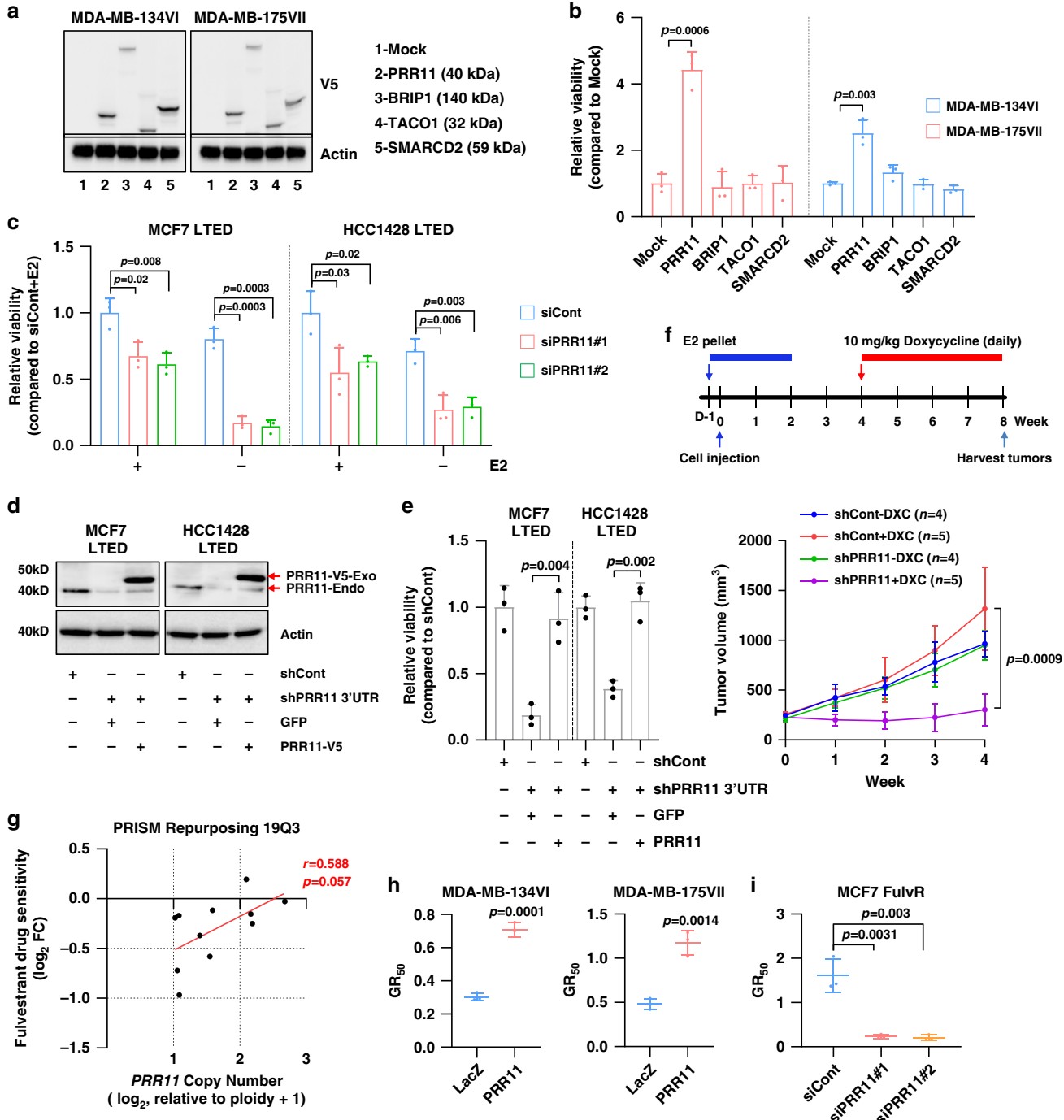

**Fig. 3 PRR11 overexpression confers resistance to antiestrogens. a** Lysates of MDA-MB-134VI and MDA-MB-175VII cells that had been transduced with pLX304-*PRR11*, -*BRIP1*, -*TACO1*, and -*SMARCD2* were subjected to immunoblot analysis. **b** Low density monolayers of MDA-MB-134VI and MDA-MB-175VII pLX304-*PRR11*, -*BRIP1*, -*TACO1*, and -*SMARCD2* cells were grown in estrogen-deprived condition. After 2 weeks, cell monolayers were stained with crystal violet and cell viability quantified as described in Methods. Data represent the mean ± SD of three replicates (two-tailed unpaired *t*-tests). **c** MCF7 LTED and HCC1428 LTED cells were transfected with *PRR11* siRNAs. Low density monolayers of cells were treated ± 1 nM estrogen (E2) for 10 days. Cell monolayers were stained with crystal violet. Data represent the mean ± SD of three replicates (two-tailed unpaired *t*-tests). **d**, **e** MCF7 LTED and HCC1428 LTED cells were transduced with shRNA targeting the 3′ UTR of *PRR11* and then, re-transduced with pLX304-*GFP* or pLX304-*PRR11*. Cell lysates were subjected to immunoblot analysis (**d**). Upper and lower arrows indicate exogenous and endogenous PRR11, respectively. Low density monolayers of cells shown in **d** were grown in absence of E2 for 10 days (**e**). Data represent the mean ± SD of three replicates (two-tailed unpaired *t*-tests). **f** MCF7 cells stably expressing doxycycline-inducible-*PRR11* shRNA and control shRNA were injected s.c. in the dorsum of athymic ovariectomized mice supplemented with a 14-day release 17β-estradiol pellet. After 4 weeks, mice were randomized to treatment with 10 mg/kg doxycycline (i.p) for 4 weeks. Each data point represents the mean tumor volume in mm³ ± SD; *n* of mice per group are shown in parenthesis (two-tailed unpaired *t*-tests). **g** Fulvestrant sensitivity of ER⁺ breast cancer cell lines (*n* = 11, PRISM Repurposing 19Q3 dataset). *Y*-axis, drug sensitivity, represents relative barcode abundance following fulvestrant treatment (Pearson correlation). **h**, **i** Fulvestrant GR₅₀ were calculated using the GR metrics calculator[56] (http://www.grcalculator.org/grcalculator/). Cell numbers on days 0 and 6 were used as the input data. MDA-MB-134VI and MDA-MB-175VII cells were stably transduced with pLX302-*LacZ* (control) or pLX302-*PRR11* (**h**). MCF7 FulvR cells were transfected with control or *PRR11* siRNA (**i**). Data represent the mean ± SD of three replicates (two-tailed unpaired *t*-tests). Source data are provided as a Source data file.

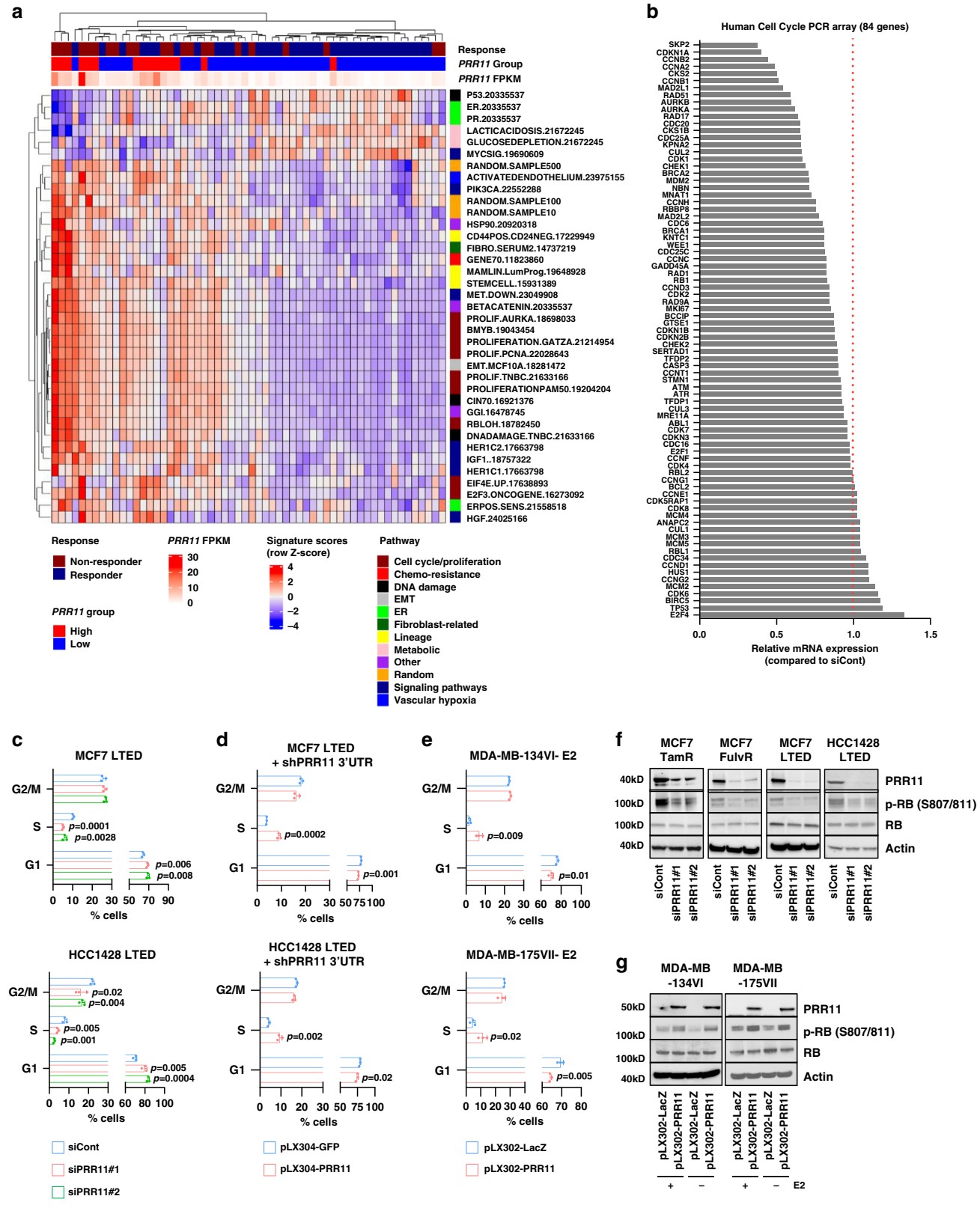

whether PRR11 affects phosphatidylinositol-3,4,5-trisphosphate (PIP3) formation, we utilized live cell imaging with a GFP-based biosensor fused with the PH domain of AKT. Detection of this biosensor at the plasma membrane is a surrogate of PI3K-induced PIP3 formation. Indeed, *PRR11* silencing by siRNA resulted in a decrease of GFP-biosensor signals at the plasma membrane and

membrane ruffling in MCF7 LTED cells (Supplementary Movies 1 and 2; control siRNA and siRNA targeting *PRR11*, respectively).

Proline-rich (PR) motifs bind to src homology 3 (SH3) domains, which are critical for the assembly of signaling complexes involved in aberrant cell proliferation[35]. Of potential

**Fig. 4 PRR11 overexpression enhances cancer cell proliferation. a** Single sample gene set analysis was performed using a set of 125 previously reported breast cancer-related signatures. Gene sets that were differentially enriched between *PRR11* high vs. low tumors (FDR < 0.01). **b** Complementary DNA (cDNA) of MCF7 LTED cells transfected with control or *PRR11* siRNA was tested in a 84-cell cycle gene PCR array. Expression of 6 genes in the array was reduced by *PRR11* siRNA transfection (FC < 0.5). Each data point represents the average of duplicate experiments. **c** MCF7 LTED and HCC1428 LTED cells transfected with control or *PRR11* siRNA for 48 h were stained with propidium iodide and analyzed by flow cytometry. Data represent the mean ± SD of three replicates (two-tailed unpaired *t*-tests). **d** MCF7 LTED and HCC1428 LTED cells transduced with shRNA targeting the 3′ UTR of *PRR11* and either pLX304-*GFP* or pLX304-*PRR11* were stained with propidium iodide and analyzed by flow cytometry. Data represent the mean ± SD of three replicates (two-tailed unpaired *t*-tests). **e** MDA-MB-134VI and MDA-MB-175VII cells transduced with pLX302-*LacZ* or pLX302-*PRR11* were grown in estrogen (E2)-deprived condition for 4 days. Cells were stained with propidium iodide and analyzed by flow cytometry. Data represent the mean ± SD of three replicates (two-tailed unpaired *t*-tests). **f** HCC1428 LTED, MCF7 LTED, MCF7 TamR, and MCF7 FulvR cells were transfected with control or *PRR11* siRNA for 48 h. Cell lysates were subjected to immunoblot analysis. **g** MDA-MB-134VI and MDA-MB-175VII cells transduced with pLX302-*LacZ* or pLX302-*PRR11* were grown ± 1 nM E2 for 4 days. Cell lysates were subjected to immunoblot analysis. Source data are provided as a Source data file.

relevance to PRR11, the p85 regulatory subunit of PI3K contains a SH3 domain and stabilizes the p110α catalytic subunit of PI3K[36]. Thus, to explore if PRR11 would interact with the PI3K pathway, we examined whether PRR11 associates with p85 via its PR motif. In MCF7 and HCC1428 cells, PRR11 was physically associated with p85α, as measured by both co-immunoprecipitation followed by immunoblot analysis and proximity ligation assay (PLA; Fig. 5c, d). This association between PRR11 and p85α was confirmed in HEK293 cells transfected with exogenous PRR11 and Flag-tagged p85α (Fig. 5d and Supplementary Fig. 6a).

The intermolecular interaction between the SH3 and PR domains in p85 monomers mediates p85 homodimerization[37,38]. The p85 homodimer contains four SH2 domains and, as a consequence, outcompetes p85/p110 heterodimers for binding to phosphorylated Tyr residues in insulin receptor substrate 1 (IRS1), thus inhibiting insulin/IGF-stimulated PI3K activity[39]. Therefore, we hypothesized that the association of surplus PRR11 with p85α would impair the formation of p85 homodimers (Fig. 5e). This would potentially enhance both the association of heterodimeric p85/p110 with IRS1 at the plasma membrane and PI3K activation stimulated by insulin/IGFs. To test this hypothesis, we co-transduced *PIK3R1* (p85α) tagged with either Flag or human influenza hemagglutinin (HA) into HEK293 cells. Ectopic expression of *PRR11* impaired the association between p85α-Flag and p85α-HA as measured by HA and Flag precipitation followed by Flag and HA immunoblot, respectively (Fig. 5f), and by PLA (Supplementary Fig. 6b), suggesting PRR11 inhibits p85 homo-dimerization. In MCF7 LTED cells stably expressing both *PIK3R1*-Flag and doxycycline-inducible *PIK3R1*-HA, knockdown of *PRR11* resulted in an increase of the p85 homodimers (Fig. 5g and Supplementary Fig. 6c). *PRR11* silencing also reduced insulin-mediated association of p110α with IRS1 and insulin/IGF-stimulated p-AKT in MCF7 LTED and HCC1428 LTED cells (Fig. 5h, j and Supplementary Fig. 6d). Conversely, *PRR11* overexpression enhanced the p110α-IRS1 association and p-AKT in MDA-MB-175VII cells (Fig. 5i, j).

To determine whether the PR domain in PRR11 is crucial to the interaction of PRR11 with p85, we generated *PRR11* mutant (*PRR11*-ΔPR) lacking the PR motif as revealed by Motif Scan, an in silico motif prediction analysis (Supplementary Fig. 7a, b). We co-transfected Flag-tagged *PIK3R1* with either V5-*PRR11*-WT or V5-*PRR11*-ΔPR into HEK293 cells. Immunoprecipitation with a Flag antibody revealed that deletion of the PR motif in PRR11 reduced its interaction with p85α compared to PRR11 WT (Fig. 5k), suggesting that the PR motif of PRR11 mediates the association with p85α. Moreover, PRR11-ΔPR did not induce p-AKT levels to the same degree as PRR11 wild type (WT) in MCF7 LTED and HEK293 cells stably transduced with *PRR11* 3′ UTR shRNA (Supplementary Fig. 7c). Together, these data suggest that the PR motif of PRR11 promotes association with

p85α, leading to reduced p85 homodimerization and enhanced PI3K activation.

***PRR11* amplification is associated with PI3K activation**. We next computed signature scores of gene sets associated with the insulin/PI3K pathway in ER⁺/HER2⁻ breast tumors in METABRIC. Signature scores of these gene sets were significantly higher in tumors harboring *PRR11* gain or amplification compared to those with *PRR11* deletion or diploid tumors (Fig. 6a, b). Similarly, hallmark gene sets associated with the PI3K/AKT pathway, such as "PI3K_AKT_MTOR_SIGNALING" and "MTORC1_SIGNALING", were enriched in METABRIC ER⁺/HER2⁻ breast tumors with *PRR11* gain or amplification compared to *PRR11* deleted/diploid tumors (Fig. 6c). We then computed the connectivity map (CMap) scores[40] with the list of genes significantly upregulated in ER⁺/HER2⁻ breast cancers with *PRR11* gain or amplification vs. diploid/deletion in METABRIC, and in ER⁺ breast cancers with *PRR11* amplification vs. no amplification in TCGA (Top 150 genes, FDR < 0.05). PI3K inhibitors and AKT signaling loss-of-function (LOF) were found as perturbation classes with < −95 connectivity score (*tau*) in MCF7 cells (Fig. 6d), representing an opposite connectivity between perturbation and gene set. This suggests that genes overexpressed in *PRR11* amplified ER⁺ breast cancers can be downregulated by perturbations that inhibit PI3K/AKT. Of note, CDK4/6 inhibitors exhibited similar opposite connectivity, suggesting these agents may also be effective against *PRR11*-amplified cancers.

Finally, in the METABRIC and TCGA datasets, *PRR11* amplification and *PIK3CA* mutations were mutually exclusive of each other in ER⁺ breast cancers. This mutual exclusivity also supports the notion that *PRR11* amplification is functionally linked to aberrant activation of PI3K (Fig. 6e).

***PRR11*-amplified breast cancer cells rely on the PI3K pathway**. The data shown so far suggest that a potential oncogenic role of PRR11 depends on activation of PI3K and, as such, PI3K inhibitors would be effective against *PRR11*-overexpressing breast cancer cells. To explore this, we first utilized MCF10A cells that require EGF and insulin to propagate. *PIK3CA* knockdown significantly inhibited the proliferation of MCF10A cells grown in media containing EGF/insulin or insulin alone (Supplementary Fig. 8a, b). *PIK3CA* knockdown abolished growth promoted by *PRR11* overexpression, suggesting that *PRR11*-mediated cell growth requires *PIK3CA*. Consistent with these data, the *PIK3CA* dependence score of 57 breast cancer cell lines significantly correlated with *PRR11* copy number in DEMETER2, a combined large-scale RNAi screening dataset[41] (Fig. 7a). In the PRISM repurposing 19Q3 dataset[30], *PRR11* copy number of 27 breast cancer cell lines significantly correlated with sensitivity to the

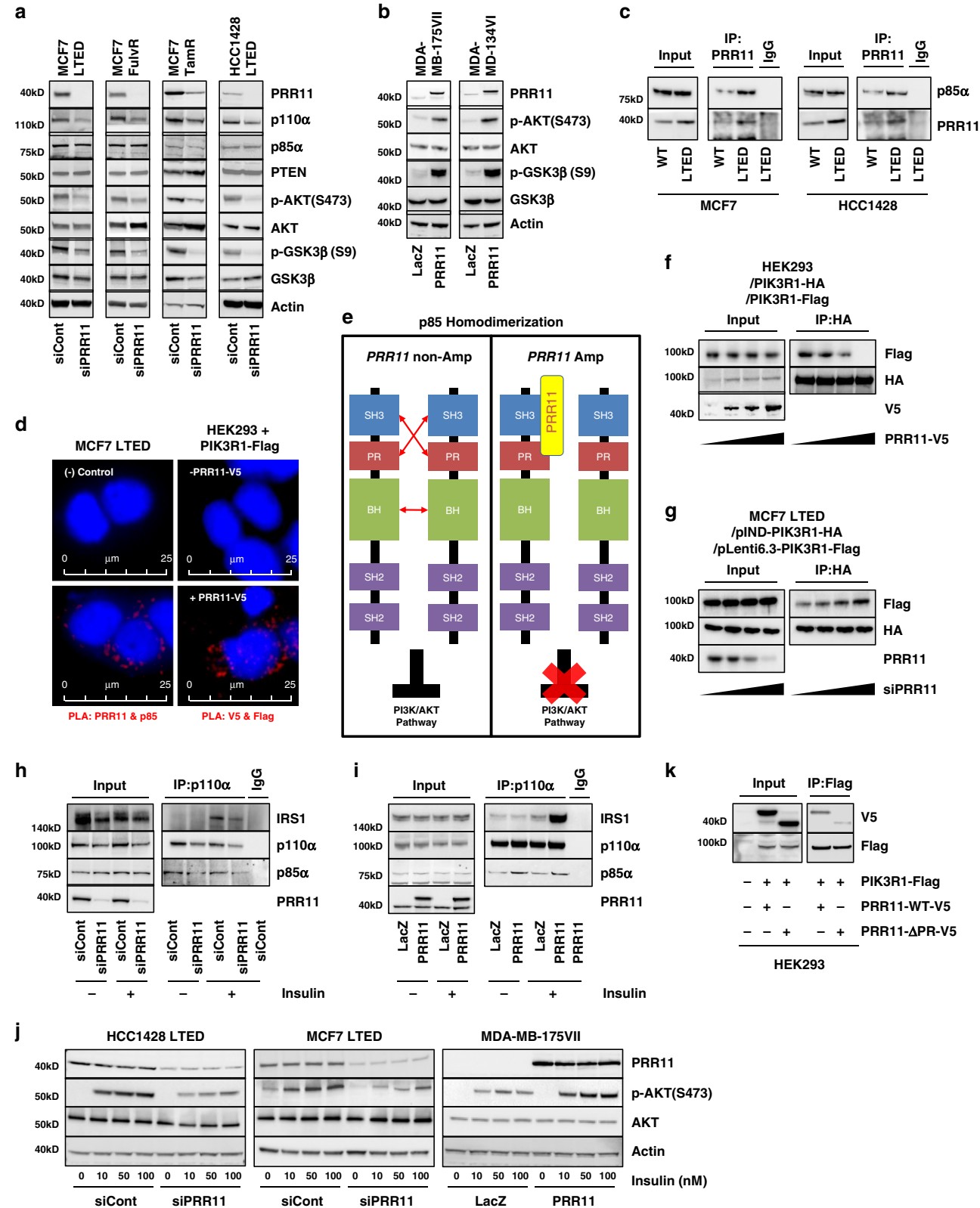

PI3K inhibitors pictilisib and taselisib (Fig. 7b). Furthermore, breast cancer cell lines with *PRR11* amplification displayed significantly higher sensitivity to the PI3K inhibitor pictilisib compared to cell lines without *PRR11* amplification in the LINCS MGH/Sanger dataset of Drug/Cell-line Browser (DCB[42]; Fig. 7c). It was previously shown that ectopic expression of mutant *PIK3CA* sensitizes these cells to PI3K inhibitors[43]. Likewise,

*PRR11* overexpression sensitized MCF10A and MDA-MB-134VI cells to PI3K inhibitors (Fig. 7d, e and Supplementary Fig. 8c, d). Together, these suggest that *PRR11*-overexpressing cells rely on *PIK3CA* and, as a result, are highly sensitive to PI3K inhibitors.

We finally tested whether genetic or pharmacological inhibition of PI3K would overcome PRR11-mediated resistance to estrogen suppression. *PIK3CA* knockdown with siRNA

**Fig. 5 PRR11 overexpression reduces p85 homodimers and enhances ligand-induced PI3K activation. a** Lysates of MCF7 LTED, FulvR, TamR, and HCC1428 LTED cells transfected with *PRR11* or control siRNA for 48 h were subjected to immunoblot analysis. **b** Lysates of MDA-MB-175VII and MDA-MB-134VI cells transduced with pLX302-*LacZ* or pLX302-*PRR11* were subjected to immunoblot analysis. **c** MCF7 parental, MCF7 LTED, HCC1428 parental, and HCC1428 LTED cell lysates were immunoprecipitated with PRR11 or IgG antibodies. Immune complexes were then subjected to immunoblot analysis. **d** MCF7 LTED cells and HEK293 cells transduced with pLenti7.3-*PIK3R1*-Flag and pLX302-*PRR11*-V5 were subjected to proximity ligation assay (PLA) with PRR11, p85α, V5 and Flag antibodies. **e** Schema of p85 monomers associating via a SH3-PR domain intermolecular interaction, potentially disrupted by *PRR11* overexpression. **f, g** HEK293 cells were co-transduced with pLenti7.3-*PIK3R1*-Flag, -*PIK3R1*-HA, and pLX302-*PRR11*-V5 (0, 0.25, 0.5, 1 μg; **f**). MCF7 LTED cells that had been stably transduced with pIND-*PIK3R1*-HA and pLenti6.3-*PIK3R1*-Flag were transfected with *PRR11* siRNA (0, 1, 5, 25 pM) for 48 h in presence of 2 μg/mL doxycycline (**g**). Cell lysates were precipitated with HA or Flag antibodies and then subjected to immunoblot analysis. **h, i** MCF7 LTED cells transfected with control or *PRR11* siRNA (**h**) and MDA-MB-175VII cells transduced with pLX302-*LacZ* or pLX302-*PRR11* (**i**) were serum-starved for 24 h and then treated with 100 nM insulin for 10 min. Cell lysates were prepared and immunoprecipitated with a p110α antibody or IgG. Antibody pulldowns were then subjected to immunoblot analysis. **j** MDA-MB-175VII cells transduced with pLX302-*LacZ* or pLX302-*PRR11* were serum-starved for 24 h and then treated with 100 nM insulin for 10 min. MCF7 LTED and HCC1428 LTED cells transfected with control or *PRR11* siRNA were treated with insulin in the same way. Cell lysates were subjected to immunoblot analysis. **k** HEK293 cells were co-transduced with pLenti7.3-*PIK3R1*-Flag and either pLX302-*PRR11* wild type (WT) or pLX302-*PRR11* ΔPR, a mutant lacking the PR motif. Lysates were prepared and immunoprecipitated with a Flag antibody; immune complexes were then subjected to immunoblot analysis. Source data are provided as a Source data file.

significantly abrogated estrogen-independent growth promoted by *PRR11* overexpression in MDA-MB-134VI cells (Fig. 7f and Supplementary Fig. 8e). In addition, treatment with the PI3K inhibitors alpelisib and taselisib abolished estrogen-independent growth of MDA-MB-175VII and MDA-MB-134VI cells stably transduced with *PRR11* (Fig. 7g, h). Consistently, PI3K inhibitors led to complete blockade of cyclin D1 protein levels that were induced by *PRR11* overexpression (Supplementary Fig. 8f). These data suggest that PRR11-mediated escape from estrogen suppression can be blocked by PI3K inhibition.

## Discussion

The 17q23 locus is amplified in ≈7% of breast cancers and has been suggested as a molecular subgroup through a clustering analysis of joint copy number and gene expression[15]. This cluster (IntClust1), which is predominantly composed of highly proliferative ER[+] luminal B breast cancers, exhibits a poor prognosis and high genomic instability. A recent report showed that IntClust1 is one of four clusters associated with a high risk of distant relapse of ER[+] breast cancers in METABRIC[16]. Initial studies aimed at identifying potential oncogenes in 17q23 focused on genes in this locus that are both amplified and overexpressed. *RPS6KB1* and *TBX2* were first proposed as putative candidates following extensive mapping of the amplicon in breast tumors and breast cancer cell lines[44]. Subsequent comprehensive analysis of copy number and gene expression predicted *MUL*, *APPBP2*, and *TRAP240* as potential oncogenes[27]. However, functional studies of these alterations have been incomplete. More recently, *WIP1* (*PPM1D*) and *MIR21*, also located in 17q23, were shown to cause resistance to anti-HER2 therapy[45]. However, *WIP1* and *MIR21* were not associated with a worse outcome of patients with ER[+] breast cancer treated with endocrine therapy in the Kaplan–Meier Plotter dataset (data not shown). In this report, our comprehensive analysis with survival data of patients with ER[+] breast cancer suggests that *PRR11* is strongly associated with breast cancer progression.

Aberrant activation of PI3K/AKT/β-catenin has been suggested as a mechanism by which PRR11 promotes cell proliferation in ovarian and hepatocellular carcinoma[19,46], but the molecular basis for a potential role of PRR11 in cancer virulence is unclear. We show herein that the PR domain of PRR11 physically binds the SH3 domain of p85, thus inhibiting p85 homodimer formation. Homodimers or monomers of p85 form a sequestration complex with IRS1, thus competing with p85-p110 dimers for binding to IRS1 at the plasma membrane and attenuating insulin/IGF stimulated PI3K activity[39]. Hence, an excess of PRR11 would

favor p85-p110 dimers and be permissive of p110 binding to IRS1, retention of p110 at the plasma membrane, and enhanced PIP3 formation. Homodimers of p85 can also inhibit PI3K signaling via direct association and stabilization of PTEN[37], but we did not observe that knockdown of *PRR11* alters PTEN protein levels (Fig. 5a).

We and others have identified hyperactivation of the PI3K pathway as a mechanism for ER[+] breast cancers to bypass hormone dependence[9,47]. In this study, we found *PRR11* overexpression was strongly associated with resistance to estrogen suppression in primary ER[+] postmenopausal tumors treated with letrozole. Other analyses shown herein also support a functional connection between PRR11 and the PI3K pathway. For instance, there was a significant correlation between PI3K/IGF1 gene set signature scores and high *PRR11* mRNA levels in clinical cohorts of ER[+] breast tumors treated with neoadjuvant letrozole. Further, *PRR11* amplifications were mutually exclusive of *PIK3CA* mutations in genomic breast cancer databases. Next, overexpression of *PRR11* in breast epithelial and breast cancer cells stimulated growth and this effect was abolished by RNA interference of p110α and by treatment with PI3K inhibitors. Finally, *PRR11*-amplified breast cancer cell lines exhibited higher sensitivity to PI3K inhibitors compared to cells that do not harbor *PRR11* amplification. Treatment of patients with advanced ER[+] breast cancer with PI3Kα inhibitors in combination with antiestrogens has significantly improved progression-free survival[13,48]. In these studies, *PIK3CA* mutations in tumors predicted clinical benefit from the PI3K inhibitor. However, some patients in these trials with wild type *PIK3CA* also benefitted clinically, suggesting that other alterations resulting in PI3K pathway dependence also respond to PI3K inhibitors and, as such, should be explored as biomarkers for enrollment of patients into trials with this class of drugs. We posit *PRR11* amplification may also serve as a predictive biomarker of sensitivity to PI3Kα inhibitors, particularly in *PIK3CA*-wild type tumors.

In summary, we identified *PRR11*, a gene in the 17q23 amplicon, as a potential driver of antiestrogen resistance in ER[+] breast cancer. Integrative analyses, including clinical data from patients with ER[+] breast cancer treated with an aromatase inhibitor strongly implicated a role for PRR11 in endocrine resistance. PRR11 blocks p85 homodimerization and sensitizes to ligand-induced PI3K activation, suggesting that *PRR11* amplification confers resistance to estrogen deprivation through hyperactivation of the PI3K pathway. Finally, we propose that, in conjunction with endocrine therapy, PI3K may be an actionable target in ER[+] breast cancers harboring *PRR11* amplification.

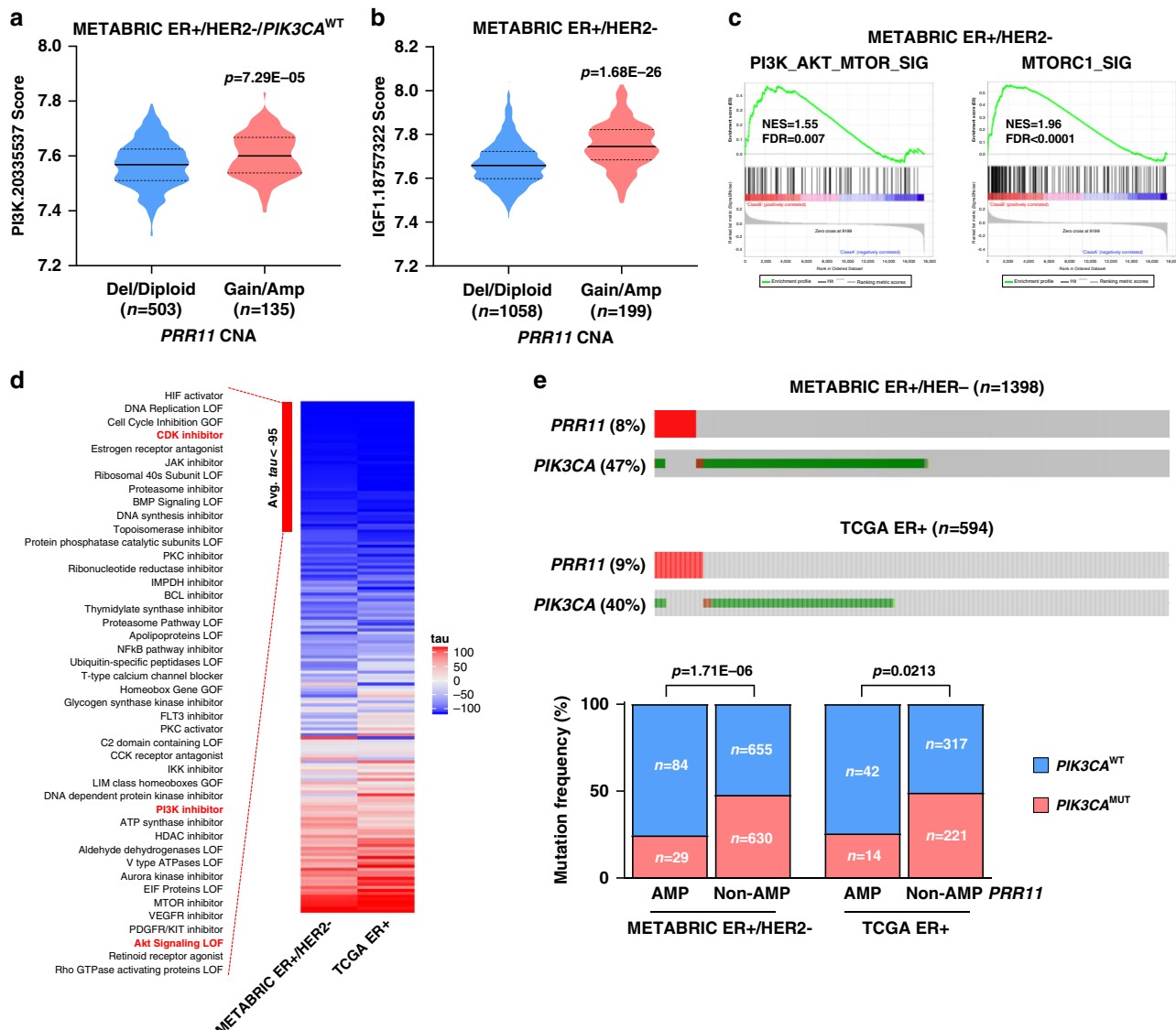

**Fig. 6 *PRR11* amplification is associated with hyperactivation of the PI3K pathway in ER$^+$ breast cancers. a**, **b** Signature score of the PI3K gene set in ER$^+$/HER2$^-$/*PIK3CA* wild type breast cancers in METABRIC plotted as a function of *PRR11* copy number (**a**: $n = 503$ and 135 for deletion/diploid and gain/amplification group, respectively). Signature score of IGF1 gene set in ER$^+$/HER2$^-$ breast cancers in METABRIC plotted as a function of *PRR11* copy number (**b**: $n = 1058$ and 199 for deletion/diploid and gain/amplification group, respectively). Data represent the mean ± SD (two-tailed unpaired *t*-tests). **c** GSEA of mRNA expression data from ER$^+$/HER2$^-$ tumors in METABRIC (*PRR11* gain/amplification vs deletion/diploid); analyses show enrichment in PI3K_AKT_MTOR_SIG and MTORC1_SIG signatures. **d** Connectivity scores (*tau*) were computed using the connectivity map (CMap) with genes significantly upregulated in ER$^+$ (TCGA) breast cancers harboring *PRR11* amplification vs. no amplification and ER$^+$/HER2$^-$ (METABRIC) breast cancers with *PRR11* gain/amplification vs. *PRR11* deletion/diploid. Connectivity score of 44 perturbation classes out of 171 are highlighted (*tau* < –95). **e** *PIK3CA* mutation frequency in ER$^+$/HER2$^-$ (METABRIC: $n = 1398$) and ER$^+$ (TCGA: $n = 594$) breast cancers plotted as a function of *PRR11* copy number alterations vs. no alterations (two-tailed Fisher's exact test). Source data are provided as a Source data file.

## Methods

**Cell lines**. The MCF7 (ATCC® HTB-22), HCC1428 (ATCC® CRL-2327), MDA-MB-175-VII (ATCC® HTB-25) and MDA-MB-134VI (ATCC® HTB-23), BT-474 (ATCC® HTB-20) human breast cancer cells, HEK293 (ATCC® CRL-1573) human embryonic kidney cells and MCF10A (ATCC® CRL-10317) breast epithelial cells were purchased from ATCC in 2018 or 2019. The 293FT (R70007) cells were purchased from Invitrogen in 2016. HCC38 and MDA-MB-231 human breast cancer cells were kindly provided by Dr. Jennifer A. Pietenpol at Vanderbilt University Medical Center. Cell lines were authenticated by the short-tandem repeat (STR) method and tested for Mycoplasma contamination. MCF7, MDA-MB-175-VII, MDA-MB-134VI, BT-474, MDA-MB-231, HEK293, and 293FT cells were maintained in DMEM/10% fetal bovine serum (FBS)/1% Antibiotic-Antimycotic (AA). HCC38 and HCC1428 cells were maintained in RPMI 1640/10% FBS/1% AA. MCF10A cells were maintained in DMEM/F12 supplemented with 5% horse serum, 20 ng/mL EGF, 10 μg/mL insulin, 0.5 μg/mL hydrocortisone, 0.1 μg/mL cholera toxin, and 1% AA. Long-term estrogen-deprived (LTED) cell

lines have been described previously[9]. To generate fulvestrant-resistant MCF7 cells, cells were cultured in the presence of increasing concentrations of fulvestrant starting at 50 nM. Cells were deemed resistant when they grew as parental cells in 1 μM fulvestrant. To generate tamoxifen-resistant MCF7 and HCC1428 cells, cells were cultured in the presence of increasing concentrations of tamoxifen starting at 500 nM. Cells were deemed resistant when they grew as parental cells in 2 μM tamoxifen. For experiments outlined here, resistant cells were removed from each drug for at least 24 h prior to treatment. MCF7 cells (29C-1) whose *PIK3CA* E545K allele has been corrected to a wild type sequence by somatic cell gene targeting were kindly provided by Dr. Ben Ho Park at Vanderbilt University Medical Center.

**Xenograft studies**. All mice were maintained according to the guidelines of the Care and Use of Laboratory Animals published by the US National Institutes of Health and the Institutional Animal Care and Use Committee. All procedures were approved by the Institutional Ethics Review Committee of the University of Texas

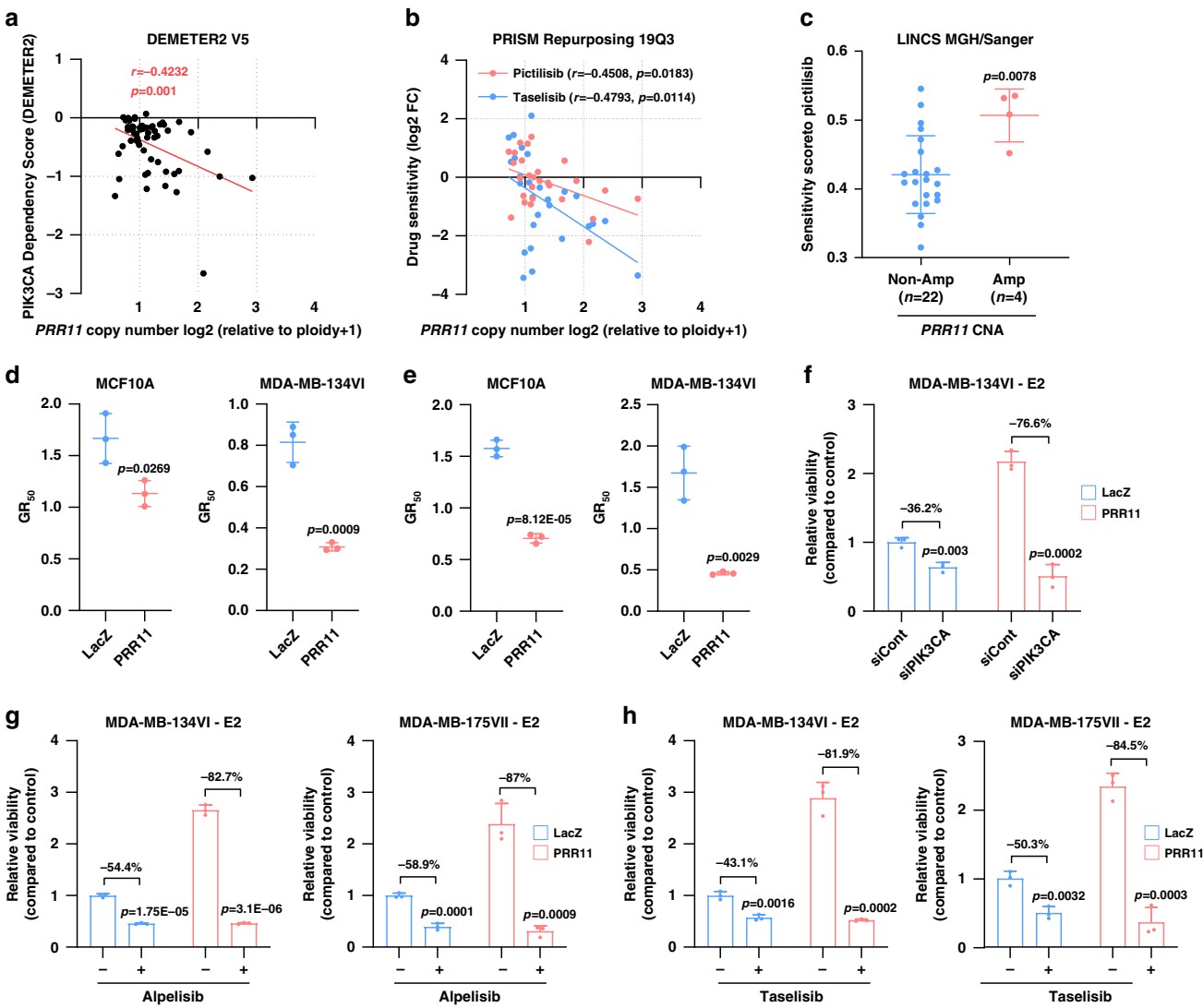

**Fig. 7 PRR11-amplified breast cancer cells are dependent on the PI3K pathway. a** PIK3CA dependency scores of 57 breast cancer cell lines were plotted against PRR11 copy number (DEMETER2 V5 dataset; Pearson correlation). **b** Sensitivity of 27 breast cancer cell lines to pictilisib and taselisib was plotted against PRR11 copy number (PRISM Repurposing 19Q3 dataset; Pearson correlation). Y-axis shows the log 2 cell fraction as per the relative barcode abundance following drug treatment. **c** Pictilisib sensitivity score of 26 breast cancer cell lines in the LINCS MGH/Sanger dataset ($n = 22$ and 4 for non-amplification and amplification group, respectively). Data represent the mean ± SD (two-tailed unpaired t-tests). **d, e** Alpelisib (**d**) and taselisib (**e**) $GR_{50}$ of MDA-MB-134VI and MDA-MB-175VII cells transduced with pLX302-LacZ or pLX302-PRR11 were calculated using the GR metrics calculator. Cell numbers counted at day 0 and day 6 were used as the input data. Data represent the mean ± SD of three replicates (two-tailed unpaired t-tests). **f** Low density monolayers of MDA-MB-134VI pLX302-LacZ and -PRR11 cells were grown in estrogen (E2)-free medium. After 14 days, cell monolayers were stained with crystal violet. Data represent the mean ± SD of three replicates (two-tailed unpaired t-tests). **g, h** Low density monolayers of MDA-MB-175VII and MDA-MB-134VI cells transduced with pLX302-PRR11 or pLX302-LacZ were treated ± 1 μM alpelisib (**g**) or ± 1 μM taselisib (**h**) in estrogen-free medium. After 14 days, cell monolayers were stained with crystal violet. Data represent the mean ± SD of three replicates (two-tailed unpaired t-tests). Source data are provided as a Source data file.

Southwestern Medical Center. Eight-week old female ovariectomized athymic mice (Hsd:Athymic Nude-Foxn1[nu], Envigo) were implanted with a 14-day release 17β-estradiol pellet (0.17 mg, Innovative Research of America). The following day, $10^7$ MCF7 cells stably expressing a doxycycline-inducible control or PRR11 shRNA were suspended in IMEM and growth factor reduced Matrigel (BD Biosciences) at a 1:1 ratio and then injected subcutaneously into the right flank of each mouse. Approximately 4 weeks later, mice bearing tumors measuring ≥250 mm³ were randomized to treatment with vehicle (0.9% NaCl) or doxycycline (10 mg/kg/daily, by intraperitoneal injection). Tumor diameters were measured with calipers weekly and tumor volume was calculated with the formula: volume = width² × length/2. After 4 weeks, tumors were harvested and homogenized using TissueLyser II (Qiagen) for subsequent immunoblot analysis.

**Gene set signature analyses.** Single-sample gene set enrichment for 125 previously published breast cancer-related gene expression signatures were computed

as described previously[17]; signatures with false discovery rate (FDR) < 0.01 were considered as differentially activated pathways between PRR11 high and low tumors. Gene set enrichment analysis (GSEA) was conducted with the javaGSEA interface downloaded from Broad Institute (http://software.broadinstitute.org/gsea/index.jsp). The h.all.v6.2.ymbols.gmt [Hallmarks] was used as gene sets database[49].

**Immunohistochemistry (IHC).** Formalin-fixed paraffin-embedded (FFPE) 4-μm tumor sections were deparaffinized. Antigen retrieval was performed with a citrate buffer (pH 6) in a decloaking chamber (Biocare). Endogen peroxidase was blocked with 3% $H_2O_2$ and protein block (Agilent). Tumor sections were next incubated with a PRR11 antibody (Novus, NBP1-83784; dilution 1:200) overnight at 4 °C. Envision (Agilent) was used for visualization with DAB as the chromogen (Agilent); hematoxylin was applied as the counterstain. Whole sections were digitally acquired using an AxioScan Z1 slide scanner (Carl Zeiss) at ×20. Automated semi-quantitative scoring was performed using QuPath software[50]. Color deconvolution

stains were set form a representative area. A cell segmentation was determined on hematoxylin OD. An object classification was trained to differentiate tumor from stroma. Percentage of PRR11+ cells was calculated with the cell detection algorithm according to the cytoplasm DAB OD mean. Each selected region was visually assessed for correct performance of the quantification algorithm excluding areas of non-invasive tumor.

**Fluorescence in situ hybridization**. Four-μm tissue sections were mounted on charged slides and hybridized overnight with a PRR11 FISH probe (EMPIRE GENOMICS) and a centromere 17 control probe (EMPIRE GENOMICS); FISH was performed as described previously[14]. Twenty to sixty tumor cell nuclei from random areas were individually evaluated with the 100× oil immersion objective by counting green PRR11 and orange centromere 17 (CEN17) signals. A PRR11/ CEN17 ratio ≥2.0 was considered as PRR11 amplification.

**siRNA transfection**. Cells seeded in 6-well plates or 60-mm dishes were transfected with 20 pmole or 40 pmole of siRNAs, respectively, using Lipofectamine RNAiMAX reagent (Invitrogen) as per the manufacturer's instructions. Control siRNA (4390843), PRR11 siRNAs (siRNA#1: 4392420-S31473, siRNA#2: 4392420-S31475), and PIK3CA siRNA (4390824-S10520) were purchased from Thermo-Fisher Scientific.

**Plasmids**. Human PRR11 open reading frame (ORF) in pENRT221 from The Ultimate™ ORF Lite human cDNA collection (Life Technologies) was cloned into pLX302 or pINDUCER20 (pIND) using Gateway™ LR Clonase™ II Enzyme Mix (ThermoFisher). To generate pLX302-PRR11-ΔPR, the PR motif was predicted using Motif Scan (https://myhits.isb-sib.ch/cgi-bin/motif_scan) and then removed from the pLX302-PRR11-WT construct using Q5-Site-Directed Mutagenesis Kit (NEW ENGLAND BioLabs). pLX304-PRR11-ΔPR was generated from pLX302-PRR11-ΔPR through BP/LR cloning. SMART doxycycline-inducible PRR11 shRNA and SMART PRR11 shRNA targeting 3'UTR was purchased from Dharmacon (V3SH11252 and V3SH11240, respectively). pLenti7.3-PIK3R1-Flag and pLenti7.3-PIK3R1-HA were kindly provided by Dr. Gordon Mills at Oregon Health Sciences University. To generate pLenti6.3-PIK3R1-Flag and pIND-PIK3R1-HA, PIK3R1-Flag and PIK3R1-HA were cloned into pDORN™221 using Gateway™ BP Clonase™ II Enzyme mix (ThermoFisher). Next, pDONR221-PIK3R1-Flag and pDONR221-PIK3R1-HA were cloned into pLenti6.3/V5-DSET™ Gateway™ vector (ThermoFisher) and pINDUCER20, respectively, using Gateway™ LR Clonase™ II Enzyme Mix. pLX304-PRR11(HsCD00444919), -BRIP1 (HsCD00440250), -SMARCD2 (HsCD00440288) and -TACO1 (HsCD00442382) were purchased from DNASU.

**Lentiviral transduction**. To generate stably transduced cell lines, 1 μg of the plasmids were co-transfected with 0.75 μg of psPAX2 (2nd generation lentiviral packaging plasmid) and 0.5 μg of pMD2.G (VSV-G envelope expressing plasmid) into 293FT cells using Lipofectamine2000 (ThermoFisher). Cell medium was changed to fresh medium 24 h post-transfection, and cells were collected 48 h later. Lentiviral particles of sgRNA editing PRR11 were purchased from Dharmacon (VSGH11937-247492131). Virus-containing medium was applied to target cells with 8 μg/mL polybrene. Puromycin were used for selection of cells transduced with SMART doxycycline-inducible PRR11 shRNA, SMART PRR11 shRNA targeting 3' UTR, pLX302-PRR11, pLX302-LacZ, control sgRNA and PRR11 sgRNA. Blasticidin were used for selection of cells transduced with pLX304-GFP, -PRR11, -BRIP1, -SMARCD2, -TACO1 or pLenti6.3-PIK3R1-Flag. MCF7 LTED cells transduced with pIND-PIK3R1-HA were selected with G418 sulfate.

**Cell proliferation assays**. After 24 h from transfection with control siRNA or PRR11 siRNAs, HCC1428, and MCF7 (LTED, TamR, and FulvR) cells were seeded in 6-well plates. Cells were trypsinized and then counted every 3 days for 6 days. Number of cells were counted with Z2 coulter counter analyzer (Beckman coulter).

**RT-qPCR**. RNA was extracted from cells using Maxwell RSC simplyRNA Cells Kit (Promega) according to the manufacturer's protocol. cDNA was synthesized using the iSCRIPT cDNA synthesis Kit (Bio-Rad) and then subjected to PCR with PowerUp™ SYBR™ Green Master Mix (ThermoFisher), PIK3CA (Qiagen), or GAPDH (Qiagen) primers using QuantStudio3 Real-Time PCR System (ThermoFisher).

**Dual luciferase assay**. Cells ($1 \times 10^4$/well) were seeded in 96-well plates in triplicate. Next day, cells were transfected with pGLB-MERE and pCMV-Renilla, each with either control or PRR11 siRNA. Twenty-four h post-transfection, cells were switched to estrogen-free medium (IMEM/5% charcoal stripped FBS) for 24 h, followed by treatment with 1 nM estradiol for 24 h. Renilla and firefly luciferase activities were measured using Dual-Luciferase® Reporter Assay System according to manufacturer's protocol (Promega).

**Clonogenic assays**. MCF7 and HCC1428 (LTED, TamR, and FulvR) cells were seed in 12-well plate and then transfected with control siRNA or PRR11 siRNA. Cells were grown ± 1 nM estradiol, ± 1 μM tamoxifen or ± 1 μM fulvestrant for 10 days. MDA-MB-175VII cells transduced with pLX304-PRR11, -BRIP1, -SMARCD2 and -TACO1 were grown in absence of estradiol for 14 days. MDA-MB-175VII cells and MDA-MB-134VI cells transduced with pLX302-PRR11 or -LacZ were grown in absence of estradiol ± 1 μM taselisib or ± 1 μM alpelisib for 14 days. MCF7 LTED cells transfected with control siRNA or PRR11 siRNA were grown ± 1 nM estradiol ± 1 μM taselisib or ± 1 μM alpelisib for 14 days. MCF10A pLX302-PRR11 or -LacZ cells were grown ± 1 μM taselisib or ± 1 μM alpelisib for 10 days. Fresh media containing drugs were replaced every 4 days. Cells were fixed with cold-iced methanol for 10 min at −20 °C and then stained with 0.5% crystal violet solution for 10 min at room temperature. Stained monolayers were imaged using Gelcount mammalian cell colony counter (Oxford Optronix). For quantification, cell monolayers were dissolved with 10% acetic acid for 15 min at room temperature. Supernatants were transferred into 96-well plates for measuring absorbance at 560 nm using GloMax Discover microplate reader (Promega).

**Immunoblot analysis**. Cells were lysed with RIPA buffer (ThermoFisher) containing protease inhibitors (Protease Inhibitor Cocktail, Roche) and phosphatase inhibitors (PhosSTOP, Roche), scraped and then incubated on ice for 30 min. Supernatants were collected after centrifugation at 14,000 rpm for 10 min. Protein concentration in cell lysates was measured with the Pierce BCA Protein Assay kit (ThermoFisher). Thirty μg of protein were subjected to SDS-PAGE followed by transfer to nitrocellulose membrane for immunoblot analysis. Information of antibodies used in this study is included as Supplementary Table 6.

**Cell cycle analysis**. After 48 h from transfection with control siRNA or PRR11 siRNAs, cells were trypsinized and then fixed with 70% ethanol for 3 h at −20 °C. Cells were washed x3 with PBS and then resuspended in PBS containing 100 μg/mL RNase A and 40 μg/mL propidium iodide for 10 min at room temperature. Stained cells were analyzed with BD LSRFORTESSA (BD Biosciences) followed by profiling of cell cycle phases with the Dean-Jett-Fox model of FlowJo (ver.10). Gating strategy is shown in Supplementary Fig. 4i.

**PCR array**. RNA was extracted from MCF7 LTED cells transfected with control siRNA or PRR11 siRNA for 48 h using Maxwell® RSC simplyRNA Cells Kit (Promega). Complementary DNA (cDNA) was synthesized using RT2 First Strand Kit according to the manufacturer's protocol and then subjected to real-time PCR using the RT² Profiler™ PCR Array Human Cell Cycle (Qiagen) in duplicate.

**Live cell imaging**. A plasmid encoding the GFP-based AKT-PH biosensor was designed as described in a previous report[51]. Briefly, the PH domain of AKT1 (residues 2-149) was fused with the COOH terminus of enhanced GFP. MCF7 LTED cells were transduced with pEGFP-AKT-PH and then sorted for GFP expression by flow cytometry. For three-dimensional imaging, we used a modified variant of the light-sheet microscope[52]. Briefly, this microscope illuminates the specimen from the epi-direction with an obliquely launched light-sheet. The beam is rapidly scanned, and subsequent fluorescence descanned, using a mirror galvanometer. For primary, secondary, and tertiary objectives, we used a high-NA silicone immersion objective (Nikon ×100 NA 1.35, 0.28–0.31 mm working distance), a high-NA air immersion objective (Nikon ×40 NA 0.95, 0.25–0.16 mm working distance), and a bespoke glass-tipped objective (AMS-AGY v1.0, NA 1.0, 0 mm working distance. https://andrewgyork.github.io/high_na_single_objective_lightsheet/), respectively. Images were acquired for each plane on a high-speed scientific CMOS camera (Hamamatsu Flash 4.0) using custom LabView-based software, sheared in the frequency-domain with Python, and rotated with an affine transform using either MATLAB or IMOD. No deconvolution was performed on the data shown. For each condition, 8 positions in a 35 mm dish were chosen at random, and each position encompassed multiple cells. Movies were acquired at a volumetric imaging rate of 0.1 Hz for 10 min.

**Immunoprecipitation**. All immunoprecipitation experiments were performed using the Dynabeads™ Protein G Immunoprecipitation Kit (ThermoFisher) according to the manufacturer's protocol. Briefly, 10 μg of antibodies were pre-incubated with 1.5 mg Dynabeads for 10 min at room temperature and then incubated overnight at 4 °C with 1 mg of cell lysate. Precipitates were eluted from the magnetic bead-antibody-antigen complex using the elution buffer containing NuPAGE LDS sample buffer (ThermoFisher) and NuPAGE sample reducing agent (ThermoFisher) for 10 min at 70 °C. Eluted samples and 20 μg of input lysates were subjected to immunoblot analysis.

**Proximity ligation assay**. MCF7 LTED ($5 \times 10^4$/well) cells were seeded in 8-well chamber slides (Lab-Tek, 177445) in triplicate. HEK293 cells ($5 \times 10^4$) were seeded 24 h post-transfection with pLX302-PRR11-V5, pLenti7.3-PIK3R1-Flag, and pLenti7.3-PIK3R1-HA. PRR11 (Cat# LS-B15222, LS Bio). p85 (Cat# 05-212, Millipore), V5 (Cat# 13202, Cell Signaling Technology), HA (Cat# 3724, Cell Signaling Technology) and Flag (Cat# 8146, Cell Signaling Technology) antibodies were used for this assay. PLA was performed with Duolink In Situ Red Starter Kit Mouse/

Rabbit (Sigma) according to the manufacturer's protocol and then imaged with a DMi8 inverted microscope (Leica). The number of PLA foci was quantified by Duolink Image Tool software; 5 images per sample were analyzed.

**Statistics and reproducibility**. Pearson $r$ correlation, hazard ratio and $t$-tests (Nonparametric tests) were performed with GraphPad Prism version 8 or Microsoft Excel 2016. Data were represented as mean ± SD. All experiments were conducted at least three times. A $p$ value < 0.05 was considered statistically significant. A false discovery rate (FDR) was computed using the Benjamini–Hochberg procedure. R version 3.5.2 and R studio version 1.1.463 were used.

**Reporting summary**. Further information on research design is available in the Nature Research Reporting Summary linked to this article.

## Data availability
All data associated with this study are present in the paper or Supplementary information. Raw RNA sequencing and clinical data of ER$^+$ breast tumors following long-term neoadjuvant treatment with letrozole were obtained from the previous report[17] and are available in the Sequence Read Archive under Bioproject accession code PRJNA605185 and have been deposited in the Gene Expression Omnibus under accession code GSE145325. RNA-seq (fragments per kilobase million; FPKM) and breast tumor Ki67 data of the cohort of Miller et al.[22] were downloaded from Supplementary information of the corresponding article. Raw RNA-seq data of the cohort of Giltnane et al.[14] was obtained from the Sequence Read Archive under Bioproject accession code PRJNA272565. These published clinical cohorts are summarized in Supplementary Table 1. Somatic mutations, normalized gene expression, and clinical data in The Cancer Genome Atlas (TCGA; Cell 2015[53]), METABRIC (Nature 2012 and Nat Commun 2016[54]) were downloaded from cBioPortal[55]. Copy number aberration and mutation data of metastatic breast cancers were obtained from The Metastatic Breast Cancer Project (https://www.mbcproject.org/), a project of Count Me In (https://joincountmein.org/). Gene expression, copy number, dependence and drug sensitivity data of breast cancer cell lines were downloaded through the DepMap portal (https://depmap.org/portal/). The code for 125 breast cancer-related signature is available at https://github.com/kmlee1982/Arteaga_lab. Source data are provided with this paper.

## Code availability
The code for 125 breast cancer-related signature is available at https://github.com/kmlee1982/Arteaga_lab.

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

## Acknowledgements

This study was supported by NCI Breast SPORE grant P50 CA098131, Vanderbilt-Ingram Cancer Center P30 CA68485, UTSW Simmons Cancer Center P30 CA142543, CPRIT RR170061 grant (C.L.A.), Susan G. Komen Breast Cancer Foundation grant SAC100013 (C.L.A.), grants from the Breast Cancer Research Foundation (C.L.A., L.C.C.), Susan G. Komen Postdoctoral Fellowship PDF17487926 (K.-M.L.), NIH grants R35 CA197588 (L.C.C.), U54 U54CA210184 (L.C.C.), and the Gray Foundation (L.C.C.). We thank Dr. Jennifer A. Pietenpol, Dr. Ben Ho Park, and Dr. Gordon Mills for providing cell lines and materials.

## Author contributions

K.-M.L., A.L.G.-Z., L.C.C., and C.L.A. designed this study. K.-M.L., A.L.G.-Z., V.M.J., and K.D. contributed to development of methodology. K.-M.L., C.C.L., P. G.-E., M.E.S., T.P. S., G.R., K.D., and R.F. contributed to acquisition of data. K.-M.L., A.B.H., A.S., D.R.S., L. F., K.D., and C.L.A. analyzed and interpreted data. K.-M.L., A.B.H., and C.L.A. wrote the paper.

## Competing interests

C.L. Arteaga has received research grants from Pfizer, Lilly, and Takeda. He holds minor stock options in Provista and Y-TRAP, and serves or has served in an advisory role to Novartis, Merck, Lilly, Daiichi Sankyo, Taiho Oncology, OrigiMed, Puma Biotechnology, Immunomedics, AstraZeneca, and Sanofi. He is a member of the Scientific Advisory Board (SAB) of the Susan G. Komen Foundation. L.C. Cantley is a founder and member of the SAB and holds equity in Agios Pharmaceuticals and Petra Pharmaceuticals, companies developing drugs for treating cancer. The laboratory of L.C. Cantley also receives funding from Petra. A.B. Hanker receives grant support from Takeda. A.L. Guerrero-Zotano has received grant support from Pfizer and travel support from Pfizer and Roche. V.M. Jansen is a current employee and shareholder of Eli Lilly and Company.
