## [Peer Review File · Nature Communications]

Point-by-point response to the referees' comments

Reviewers' comments (Bold italic):

Reviewer #1 (Remarks to the Author):

In this manuscript, Lee, K.-m. et al describe a vulnerability in estrogen receptor (ER)-positive breast cancers resistant to anti-estrogen treatment. They show that cells of some ER-positive breast cancers cells rely on the PI3K-pathway for survival and, thus, PI3K inhibitors would be a rational therapy to treat them. The mechanism behind this vulnerability depends on PRR11 and IRS1. The authors conclude that PI3K inhibition is a rational treatment for PRR11-amplified ER-positive breast cancers, particularly those with no activating PI3KCA mutations.

Acquired resistance to efficacious therapies remains one the main hurdles in the treatment of cancer. This is the case of anti-hormonal therapy against ER-positive breast cancers. Despite its efficacy, resistance arises in ~20% of patients. For some tumors the mechanisms of resistance have been identified, but form many others they are unknown. Thus, the authors address a relevant issue.

The manuscript has been preceded by different reports showing the relevance of the 17q21-23 amplification in resistance to hormonal therapy. On the other hand, the efficacy of PI3K inhibitors on some ER-positive breast cancers resistant to anti-hormonal therapy has been proposed in several reports, and has been already confirmed in the clinic. Thus, the most relevant finding of the study presented here is the role of PRR11. The large amount of data presented, part of them superfluous, makes following up the main finding quite challenging for the reader. In contrast, data that is critical is missing. For example, the mechanism proposed by the authors strongly relies on levels of PRR11. These should be enough to interact with as many p85 α regulatory subunit as necessary to promote the generation of p110-p85 α heterodimers. However, little data on levels of PRR11 is presented. Do all amplified tumors express the same levels of PRR11? When the authors refer to tumors with low and high PRR11 levels, where do they set the threshold? Are there mechanisms of overexpression of PRR11 other than gene amplification? Addressing these issues would greatly support the main conclusion of this paper.

We agree with the reviewer's point regarding the expression levels of PRR11 in tumors. However, we showed that *PRR11* copy number alterations significantly correlate with mRNA levels in primary breast tumors in the METABRIC dataset (**Rebuttal Fig. 1A**; included as **Supplementary Fig. 2b** of the revised manuscript) and TCGA dataset (**Rebuttal Fig. 1B**). This correlation is also observed in breast cancer cell lines at both the mRNA and protein level (**Rebuttal Fig. 1C,D**; included as **Supplementary Fig. 2c,d** of the revised manuscript). Taken together, these suggest that *PRR11* is overexpressed in tumors that harbor *PRR11* amplification, although the expression level is heterogeneous in *PRR11*-amplified breast tumors.

In our original analyses, patients with high or low *PRR11* expression were divided by the median because the primary goal of these analyses was to investigate differentially expressed genes and/or activated pathways as a function of *PRR11* expression. To further address this concern, we adopted the *PRR11* cut-off [fragments per kilobase of transcript per million mapped reads (FPKM)=3.93] used for TCGA breast tumors from the human protein atlas (<http://www.proteinatlas.org>; PMID: 25613900). Using this *PRR11* FPKM cut-off, we have divided tumors by *PRR11*-high vs. -low expression and reassessed their correlation with recurrence free survival and with 125 breast cancer-related signatures in the revised

manuscript (**Fig. 1c and 4a**). Overall, we made similar observations compared with the previous analyses where we used median PRR11 as the cut-off.

Our study does not exclude any potential mechanisms by which PRR11 is transcriptionally or translationally regulated. Rather, our study focuses on the impact of *PRR11* copy number amplification which is associated with mRNA overexpression. In fact, both *PRR11* copy number amplification and mRNA overexpression are associated with the enrichment of similar oncogenic gene set signatures.

We agree with the reviewer's comment regarding superfluous data. Therefore, we have removed the **Figs. 3f and 4b** and moved the **Figs. 2g, 4c-e, 5h, 7a and 7i** in the original manuscript to the Supplementary Data of the revised manuscript.

Rebuttal Fig. 1. A and B. *PRR11* mRNA levels plotted against *PRR11* copy number alterations (CNAs) in ER⁺/HER⁻ breast tumors in METABRIC (A) and ER⁺ breast tumors in TCGA (B). C. *PRR11* copy numbers plotted against *PRR11* mRNA expression in breast cancer cell lines of the CCLE dataset (Pearson r). D. Lysates from breast cancer cell lines were subjected to immunoblot analysis with PRR11 and actin antibodies. *PRR11*-amplified cell lines were highlighted as red. The intensity of PRR11 immunoblot bands was quantified using the Image Lab software (ver. 6.0, BioRad) and then plotted by *PRR11* amplification status in the right (t-test).

Additional points:

Of the 58 ER+ breast cancers used for the analysis shown in Fig. 1a, how many had the 17q23 amplification? Do any non-amplified tumors overexpress PRR11?

Unfortunately, we did not have enough tumor tissue in the previous study for this analysis. We had profiled copy number of those samples, but the data were not sufficient to call copy number amplification due to lack of normal (non-tumor) matched tissues.

The authors claim that “PRR11 has been reported as an estrogen-responsive gene”. However, the article referenced (Cancer Res 70, 3760-3770) has no reference to PRR11. This could be due to different names of PRR1 (also known as NECTIN1, among more than a dozen names). Since the results in the manuscript reviewed here could contradict those reported in the article published in Cancer Research, the authors should facilitate the reader the comparisons of these results by clarifying any possible confusion due to name ambiguity.

We apologize for the confusion. In the referenced study, the authors profiled gene expression followed by estrogen stimulation in MCF7 cells. Although *PRR11* was not directly mentioned in the paper, *PRR11* is

one of genes (n=323), which was significantly upregulated by estrogen stimulation. *PRR11* is included as an upregulated gene in Supplementary Table 3 of the article in *Cancer Research*.

What is the distance of the sequences encoding *BRIP1*, *SMARCD2* and *TACO1* genes to that encoding *PRR11*?

We have included a cytogenetic band of 17q showing the distance among those genes in **Supplementary Fig. 2a** of the revised manuscript (**Rebuttal Fig. 2**).

Rebuttal Fig. 2. Location of *PRR11*, *BRIP1*, *TACO1* and *SMARCD2* in the chromosome 17q.

Fig. 2e. Where was the V5 epitope introduced? The authors should present controls to show that the epitope does not perturb the function of the protein. How do the levels of expression achieved upon overexpression relate to the normal levels of expression?

These are commercially available constructs tagged with V5 at their C terminus. We did not test the function of tagged protein because the functions of each protein are unclear or unknown. Based on our previous experiments with pLX302 or pLX304 vectors and other publications using these vectors, the V5 epitope is not expected to affect their own protein function (PMID: 28978427, 24356096). In both MDA-MB-134VI and MDA-MB-175 cells, overexpression of *PRR11* achieved higher levels of *PRR11* protein compared to endogenous *PRR11* protein (**Rebuttal Fig. 3**).

Rebuttal Fig. 3. Lysates from MDA-MB-134VI and MDA-MB-175VII cells stably transduced with pLX302-*LacZ* or -*PRR11* were subjected to immunoblot analysis with *PRR11* and actin antibodies.

For readers outside the field, what is the meaning of the ATARiS score?

ATARiS is a computational quantification method of gene suppression phenotypes from multi-sample RNAi screens (PMID: 23269662). We have included this description in the revised manuscript.

Fig. 3b. Please, describe the different visible bands. Is the slow migrating band endogenous *PRR11*? If so, why does ectopically expressed *PRR11* migrate slower?

The upper band in the last column of **Fig. 3b** represents exogenous *PRR11* protein which is tagged with V5 epitope. Technically, it has slightly higher molecular weight compared to endogenous *PRR11* protein.

For clarity, we have added arrows indicating endogenous and exogenous PRR11 protein (**Rebuttal Fig. 4**).

Rebuttal Fig. 4. MCF7 LTED and HCC1428 LTED cells were transduced with shRNA targeting the 3' UTR of *PRR11* and then re-transduced with pLX304-*GFP* or pLX304-*PRR11-V5*. Cell lysates were subjected to immunoblot analysis with PRR11 and actin antibodies. Exo, exogenous; endo, endogenous.

Extended data Fig. 3b. The knock-out of PRR11 should completely abrogate expression of PRR11. However, there are detectable levels of expression in the data shown in extended data Fig. 3b. Have the authors analyzed a pool of cells, some not knocked-out?

Due to the essentiality and multiple copies of *PRR11* in MCF7 and HCC1428 cells, we failed to get clones with complete knockout of PRR11. Since we used the pooled population of cells with CRISPR knockout, PRR11 protein coded by a non-edited allele was also detected. Hence, in this revised manuscript, we focused on *PRR11* shRNA and siRNA to knock down *PRR11* expression.

The status of amplification of MCF7 cell, an experimental model repeatedly used in the manuscript is confusing: "MCF7 and HCC1428 cells, both of which display high PRR11 copy number". How to the levels of PR11 amplification of parental MCF7 cells compare to those of MCF7 LTED (long-term estrogen deprived), (FulvR) MCF7 cells tamoxifen-resistant (TamR) MCF7.

We thank the reviewer for this comment and apologize for the inadequate explanation. Based on the public dataset from Cancer Cell Line Encyclopedia, both MCF7 and HCC1428 parental cells harbor *PRR11* high copy number. We did not evaluate further genomic copy number increases in MCF7 and HCC1428 LTED cells, but we did observe an elevation of PRR11 protein levels in both LTED lines compared to their respective parental counterparts (**Rebuttal Fig. 5**).

Rebuttal Fig. 5. Lysates of MCF7 and HCC1428 LTED cells were subjected to immunoblot analysis with PRR11 and actin antibodies.

Fig. 4a. How do the authors define "tumors with high PRR11 mRNA"?

The median level of *PRR11* mRNA was used to define *PRR11*-high or -low tumors as described in figure legend of the original manuscript. However, as we replied above, we have split *PRR11*-high vs. -low

tumors with the FPKM cutoff adopted from the human protein atlas in the revised manuscript. For the Kaplan-Meier plotter analyses, the auto select best cut-off was used for splitting patients with high *PRR11* expression vs. low.

Fig. 4g. Does the rescue shown here correspond to the rescue of cell proliferation?

We apologize for the typo in the figure. Cells shown in **Fig. 4g** correspond to the rescue models shown in **Figs. 2b and 2c**. **Fig. 4g** and corresponding legend have been corrected accordingly.

Fig. 5j is rather unconvincing. The increase in the levels of p85alpha in immunoprecipitates of p110alpha induced by PRR11 is marginal.

Extended data Fig. 7D. The increase in the level of p-Akt is marginal.

Based on a quantification of immunoblot band intensity, the p85α levels induced by PRR11 overexpression were about 2-3 folds compared to respective controls (**Rebuttal Fig. 6A**). The intensity of p85α immunoblot bands was measured using the Image Lab software (ver. 6.0, BioRad).

Immunoblot for p-AKT in **Extended Data Fig. 7d** of the original manuscript was reassessed and replaced in the revised manuscript (**Rebuttal Fig. 6B**; included as **Supplementary Fig. 7c** of the revised manuscript).

Rebuttal Fig. 6. A. Quantification of the immunoblot band intensity in Fig. 5i of the revised manuscript. Bar graph in the right represents respective p85α immunoblot band intensities relative to the band in the first lane. B. Lysates from HEK293 cells transfected with pLX302-*PRR11* wild type or -*PRR11* ΔPR mutant plasmid were subjected to immunoblot analysis with V5, p-AKT(S473), AKT and actin antibodies.

Reviewer #2 (Remarks to the Author):

In this manuscript, Lee et al. investigate genes within the 17q23 amplicon that might causally contribute to the previously-documented association of this amplicon with poor clinical outcome in ER-positive breast cancers. Based on transcriptome data from clinical studies, the Proline rich 11 gene (PRR11) was identified as a top gene associated with a poor clinical response to estrogen inhibition in ER-positive breast tumours. Additional bioinformatic studies (Fig. 1 and Fig. 2a-d, g) strengthen the case for an association of PRR11 with clinical outcome and tumour cell proliferation markers. Cell-based studies were then undertaken to functionally explore the impact of PRR11 over/under-expression on in vitro breast cancer cell proliferation and drug sensitivity. Evidence is presented that PRR11 downregulation leads to some level of anti-proliferation/enhanced sensitivity to estrogen suppression, with overexpression having the opposite effect (Fig. 3). In a next series of analyses, PRR11 gene expression is linked to a PI3K activation signature in cancer and to enhanced PI3K activation in cell-based models. This is mechanistically supported by data to indicate that PRR11 can bind the p85α regulatory subunit of PI3K and hereby suppress p85 homodimer formation, allowing p85 instead to associate with the

p110 α catalytic subunit. This results in an increased binding of p110-p85 α heterodimers upon insulin stimulation to IRS1 and cellular activation of PI3K.

This manuscript contains a wealth of data and analysis. While the bio-informatic data provide substantial support for a clinical link with PRR11 with ER sensitivity, the cell-based studies to functionally strengthen this connection are less convincing. It is appreciated that cell-based assays exploring the function of a single gene may be inadequate to model the complexities of human cancer (or that this manuscript has not uncovered the real mechanism), illustrating the challenges to functionally validate genomic data from cancer.

The strongest conclusion of this manuscript is that, largely based on bio-informatic data and some cell-based observations, PRR11 amplification in ER+ breast cancer is a candidate genetic marker to aid patient selection for ER-targeted therapies. However, the role of the PRR11-PI3K signalling link remains unclear, and the current conclusion of increased cellular sensitivity to PI3K inhibitors lacks sufficient evidence.

Major points:

1. It is critical that all bio-informatic data/code are made available alongside extensive expansion of the methods to detail the key steps undertaken – essentially as provided to the reviewer upon additional request.

Data and code availability are stated in the reporting summary which is enclosed with the revised manuscript.

2. Generally, the manuscript currently gives the impression of selective modulation of estrogen sensitivity by PRR11, but the repertoire of tested breast cancer cell lines is very limited. Most likely, the authors initially focused on modulation of estrogen sensitivity by PRR11 expression, only to realize later in the experimental journey that sensitivity to other drugs was also affected, including inhibitors against PI3K or CDK4/6 (although the latter is implied rather than shown; lines 247-249). For example, Fig. 3a/c focus on the growth-inhibitory effect of PRR1 downregulation in MCF7 and HCC1429 long-term estrogen-deprived cell lines. The selection of this type of cell lines at this stage in the manuscript is somewhat puzzling as are the efforts of the authors to ‘forge’ a link with estrogen sensitivity, as at this stage of the work, it is not clear whether PRR1 downregulation leads to growth inhibition of any cancer cell line. Likewise, the data in Fig. 3d are used to make a statement on ‘the role of PRR11 in low estrogen conditions in vivo’.

The authors should test the impact of modulation of PRR11 expression, particularly downregulation, on the proliferation of a range of non-ER responsive breast cancer lines, especially triple negative and/or HER2+ cell lines. Based on their clinical observations on p. 4, lines 86-89, one would expect such cell lines to be insensitive to PRR11 downregulation and will thus strengthen the conclusions they make with regards to this protein’s role in ER-positive tumours only.

We thank the Reviewer for these suggestions. Our aim was not to forge a link between estrogen sensitivity and PRR11 levels, but to determine whether high PRR11 affects the response to estrogen *suppression*. The ‘low estrogen conditions in vivo’ are our attempt to try to reproduce the clinical scenario in the original clinical cohort where, in the setting of therapeutic estrogen suppression with letrozole (e.g., low estrogen *in vivo*), tumors with high *PRR11* expression exhibited resistance as suggested by both a high PEPI score and recurrence after 5 years of follow up. We employed MCF7 and HCC1428 LTED cells because 1) parental MCF7 and HCC1428 cells harbor high *PRR11* copy number, 2) although initially sensitive to estrogen suppression, these cells always adapt to experimental estrogen suppression, and 3) *PRR11* protein levels were elevated in LTED cells compared to parental cells (**Rebuttal Fig. 5**). We selected these cells, again, not to forge a link with estrogen sensitivity, but, because of their ability to adapt to estrogen suppression, to show that high *PRR11* levels can mediate escape from estrogen suppression, in line with the thesis of the paper. In this revised manuscript, we have used ER+ cell lines (MDA-MB-134VII and MDA-MB-175VI) without high *PRR11* copy number that are initially sensitive to antiestrogens and demonstrated that *PRR11* overexpression promotes estrogen-independent growth and

drives fulvestrant resistance, lending further support to our hypothesis that *PRR11* amplification, within the 17q23 amplicon, mediates escape from the estrogen suppression.

As requested by the Reviewer, we have extended our studies to other breast cancer subtypes. We ablated *PRR11* in two triple negative breast cancer (TNBC) cell lines, HCC38 and MDA-MB-231, and one HER2+ breast cancer cell line, BT474 (**Rebuttal Fig. 7A**; included as **Supplementary Fig. 4g** of the revised manuscript). HCC38 and BT474 cells harbor *PRR11* high copy number and also express high level of *PRR11* protein (**Rebuttal Fig. 7B**; included as **Supplementary Fig. 2d** of the revised manuscript). MDA-MB-231 cells express high levels of *PRR11* protein (**Rebuttal Fig. 7B**). *PRR11* knockdown with siRNA promoted cell proliferation in TNBC cells, which is discordant with the observation in ER+ breast cancer cells. In BT-474 cells, the anti-proliferative effect of *PRR11* silencing was not significant (**Rebuttal Fig. 7C**; included as **Supplementary Fig. 4h** of the revised manuscript). *PRR11* silencing modestly reduced p-AKT in MDA-MB-231 and BT474 cells (**Rebuttal Fig. 7A**). We have not been able to investigate the mechanism of the discrepancy in other subtypes of breast cancer as they would require a second study. Unfortunately, we are in no position to state that the mechanism linking PI3K signaling to cell viability is *specific* to ER+ breast cancers, as this would require many more cell lines and human tumors.

We would like to respectfully emphasize that the basis of this study was an observation in a clinical cohort of patients with ER+ breast cancer treated with estrogen suppression, not estrogen stimulation. Herein, we demonstrated that *PRR11* promotes hyperactivation of the PI3K/AKT pathway, a signaling pathway that, when aberrant, can bypass hormone dependence in ER+ breast cancer (PMID: 20530877). Many drivers of endocrine resistance activate growth factor signaling pathways in order to bypass estrogen dependence. A recent study by Razavi and colleagues reported that mutations/alterations on *ESR1*, transcription factors, the MAPK pathway, and other unknown pathways account for 18%, 9%, 13% and 60%, respectively, of endocrine resistance in ER+ breast tumors (PMID:30205045). Except for the mutations in *ESR1* and possibly transcription factor alterations, many of these potential drivers contribute to the resistance via alternative growth promoting signaling pathways rather than modulation of ER signaling *per se*. For instance, acquired mutations in HER2 drive resistance to endocrine therapy via PI3K/mTOR activation (PMID:30314968). Thus, we recognize that *PRR11* regulates cell proliferation independently of estrogen signaling *per se*. Further, the role of *PRR11* in cell proliferation and poor outcome has been previously suggested in various cancer types that are not associated with hormone receptor signaling (lung, pancreatic and gastric cancer).

Rebuttal Fig. 7. A. Lysates were obtained from HCC38, MDA-MB-231 and BT474 cells transfected with control or *PRR11* siRNA for 48 h and then subjected to immunoblot analysis with *PRR11*, p-AKT, AKT and actin antibodies. B. Lysates from breast cancer cell lines were subjected to immunoblot analysis with *PRR11* and actin antibodies. *PRR11*-amplified cell lines were highlighted as red. Red arrows indicate non-ER+ cell lines with high expression of *PRR11*. C. HCC38, MDA-MB-231 and BT474 cells were transfected with control or *PRR11* siRNA. After 24 h, 20,000 cells were seeded on 6-well plates and then cells were counted every 72 h for 6 days.

3. The conclusions of this manuscript with regards to sensitization by *PRR11* overexpression to *PI3K* inhibitor are inaccurate as currently presented. Indeed, from the data, it is clear that *PI3K* inhibition results in overall the same cell number at the end of the assay (Fig. 7g,h), irrespective of *PRR11* expression, yet the relative growth inhibition appears more impressive upon *PRR11* overexpression simply because *PRR11* proliferate more than control cells over the course of the assay. Therefore the putative “enhanced sensitivity” may simply be a consequence of lack of adjustment for growth rate (see further below). Specifically, all cell data are currently presented as ‘relative viability’, which most likely implies normalisation to the cell number of control cells at the end of the assay. These end-point assays have flaws, and in order to get proper insight into the role of *PI3K* inhibition in the cell models tested, the authors should perform growth rate (GR)50 and GRmax analyses for their *PI3K* inhibitor assays, as described in PMID:27135972 and related papers from the Sorger group. Although the authors use DCB data to support their conclusion re *PRR11*-mediated sensitisation to *PI3K* inhibitors, those data were also not normalised for growth rate and likely suffer from the same limitations as the author’s own results (see also: PMID: 28591115).

The statements that ‘*PRR11*-amplified cancers are highly sensitive to *PI3K* inhibitors and rely on *PIK3CA*’ should be removed from both the abstract and text (lines 267-269). This also applies to the title of the paragraph ‘*PRR11*-amplified breast cancer cells are dependent on the *PI3K* pathway’ (line 253). The final conclusion of the abstract (‘These data suggest *PI3K* inhibitors as a novel therapeutic strategy for *PRR11*-amplified ER+ breast cancers’) should also be toned down. At best, it could be speculated that *PRR11* amplification in ER+ breast cancer is a candidate genetic marker to aid patient selection for ERα targeted therapies.

We thank the reviewer for this insightful comment and agree that GR metrics assay will improve the accuracy of drug sensitivity assessment. To address this point, we reassessed the drug responses to fulvestrant, alpelisib and taselisib using the GR metrics calculator (**Rebuttal Fig. 8**). All assessments of drug sensitivity in **Fig. 3g, 3h, 7e** and **7f** of the original manuscript were replaced to the GR metrics (GR50 and GR value curves) in the revised manuscript. We observed a trend similar to the previous analyses. Furthermore, we wish to emphasize that data in **Fig. 7f-h** of the revised manuscript clearly show that PRR11 overexpression promotes estrogen-independent growth, which is blocked by PI3K inhibitors. Finally, we did not mean to imply that PI3K inhibitors are a novel therapeutic strategy. We have now clarified that ER+/PRR11 amplified tumors represent a novel subgroup that may benefit from the combination of a PI3K inhibitor and an antiestrogen. Accordingly, we have amended 'These data suggest PI3K inhibitors as a novel therapeutic strategy for PRR11-amplified ER+ breast cancers' to 'These data suggest ER+/PRR11-amplified breast cancers as a novel subgroup that may benefit from treatment with PI3K inhibitors.' in the abstract.

Rebuttal Fig. 8. A-H. GR and GR₅₀ values were calculated using the GR Metrics calculator (t-test). MDA-MB-134VI and MDA-MB-175VII cells stably transduced with pLX302-LacZ and -PRR11 were treated with multiple doses of fulvestrant for 6 days (A and B). MCF7 FulvR cells transfected with control or PRR11 siRNA were treated with multiple doses of fulvestrant for 6 days (C and D). MDA-MB-134VI and MCF10A cells stably transduced with pLX302-LacZ and -PRR11 were treated with multiple doses of alpelisib (E and F) or taselisib (G and H) for 6 days.

4. Figure 7a and extended Data Figure 8. It is not clear why MCF10A was chosen as a cell model for these critical experiments as it is not a breast cancer cell line (but an immortalized but not transformed cell line) and is also ER-negative. These experiments should instead be performed on genuine breast cancer cell line, both ER-positive and ER-negative.

We agree with the reviewer's concern. We utilized MCF10A cells because this is a cell line that relies on insulin for propagation and lacks a *PIK3CA* mutation. We now include data suggesting that estrogen-independent growth promoted by *PRR11* overexpression requires *PIK3CA* in MDA-MB-134VI ER+ breast cancer cells (**Fig. 7f** of the revised manuscript). We have moved the results with MCF10A cells to Supplementary Data.

5. Other than the FISH data presented in Extended data Fig 2b, the Authors should aim to show evidence for PRR11 protein overexpression in human breast cancer tissue. This protein can only have a functional impact if it is indeed overexpressed at the protein level.

We conducted PRR11 immunohistochemistry (IHC) in 175 primary ER+ breast tumors treated with neoadjuvant letrozole (**Rebuttal Fig. 9A**; included as **Fig. 1g** of the revised manuscript). Based on the distribution of PRR11 protein levels, PRR11-positive and -high tumors were classified by PRR11 >1% and >15%, respectively (**Rebuttal Fig. 9B**; included as **Supplementary Fig. 1b** of the revised manuscript). As a result, 14.3 % and 4.6% of tumors were defined as PRR11-positive and PRR11-high, respectively. Of note, tumors with poor response to letrozole exhibited higher levels of PRR11 protein, supporting the functional impact of PRR11 on endocrine therapy response (**Rebuttal Fig. 9C**; included as **Fig. 1h** of the revised manuscript).

Rebuttal Fig. 9. A. Representative PRR11 IHC images of ER+ primary breast tumors. B. Distribution of PRR11 positivity in ER+ primary breast tumors. C. Tumoral PRR11 protein levels were plotted by letrozole responses that were defined by on-treatment Ki67 levels as described in the previous report (PMID: 28794284).

Minor comments

1. A more detailed description of PPR1 should be given at the beginning of the manuscript, information on this protein now comes very late in the manuscript (lines 202-226 and Extended data Fig. 7). What is known about this protein? Even if nothing is known, this should be stated. I found it rather odd to have to read through the MS without having no information about this protein.

We apologize for the lack of clarity. We have included a brief description about PRR11 in the Introduction section of the revised manuscript as below.

“PRR11 has been implicated in poor outcome of various cancer types, but the molecular basis for this association is unclear.”

2. Extended data Fig 2a: what statistical test was performed?

As we described in Method section, t-tests (Nonparametric tests) were used for the calculation of p value.

3. Extended data Fig 2b: a negative control is missing: the Authors should also show a FISH image of a cancer without PRR11 amplification.

We have included the FISH image of a primary breast tumor without PRR11 amplification (**Rebuttal Fig. 10**; included as **Supplementary Fig. 2e** of the revised manuscript).

Rebuttal Fig. 10. Representative FISH image from a breast tumor specimen with PRR11 amplification. Magnification = 100x.

4. Page 5: line 118: ‘we listed 90 genes located in the 17q23 amplicon’: it was not clear whether these 90 are all the genes found in this amplicon? If not, how were these genes selected?

We listed all genes located in 17q23 amplicon based on the Atlas of Genetics and Cytogenetics (PMID: 23161685).

5. Extended Fig 3a, Legend: what does ‘low density monolayers’ mean here and in all other figure legends where it appears? The authors should be much more specific with regards to how their assays were performed when it comes to cell numbers, media changes and similar, as these are factors that heavily influence the reproducibility of drug testing assays and similar.

We apologize for the lack of clarity. Technically, clonogenic populations propagated from low number of cells (1,000 cells) seeded in 12-well plate (2-D) were considered as the ‘low density monolayers’. The information of media change was described in the “Clonogenic assays” section of Methods.

6. The information on antibody and other reagents need additional detail, for example for western blots, proximity ligation assay etc.

We have included information of antibodies used in this study as supplementary material.

Reviewer #3 (Remarks to the Author):

Using a comprehensive approach combining integrative genomic analyses and validation studies, the authors provide strong data to suggest PRR11 located in 17q23, is critical for conferring endocrine resistance through amplification of PI3K signaling. Interestingly, analysis of RNAi screening data from Project Achilles suggested that 17q23-amplified breast cancer cells carry a high dependency on PRR11. The results presented suggest that PRR11 interacts with the P85 subunit of PI3K and interferes with homodimerization and perhaps facilitating heterodimerization with p110 subunit. Additionally, they demonstrate that PRR11 was both necessary and sufficient to promote proliferation in endocrine resistant/estrogen suppressed ER+ breast cancer cells. Overall, the authors provide strong evidence that PRR11 is a critical mediator of endocrine resistance through regulation of the PI3K pathway. However, some of the conclusions need to be revisited and strengthened. Specific concerns and questions are listed below:

1) The authors suggest that there is a higher rate of PRR11 amplification in metastatic (data from the MBC project compared to METABRIC and TCGA) compared to ER+ primary breast tumors. Yet, their in vivo experiment only examined primary tumor growth. Does PRR11 promote tumor progression/metastasis in vivo? Either spontaneous or experimental metastasis assays could be done to address this question.

PRR11 has been associated with an invasive phenotype in hepatocellular and ovarian carcinoma (PMID: 30248355, 30165366), so investigating a role of PRR11 in breast cancer metastasis would be very interesting, but also a significant additional study. However, our study focused on the implication of PRR11 overexpression/amplification in endocrine resistance, another phenotype indicative of increased cancer virulence.

2) An in vivo study demonstrating the effects of PI3K inhibition on PRR11 amplified cells would greatly strengthen the clinical relevance of the manuscript. Additionally, would overexpression of PRR11 confer endocrine therapy resistance in vivo?

We thank to the Reviewer for these insightful suggestions and wholeheartedly share his/her wishes. Regarding the evaluation of endocrine therapy resistance promoted by PRR11 overexpression in vivo, unfortunately, we failed to find a feasible model due to limited tumorigenicity of ER+/PIK3CA^{WT}/PRR11 non-amplified breast cancer cell lines (MDA-MB-134VI, MDA-MB-175VII, CAMA1 and HCC1500). We and others have previously shown that the estrogen-independent growth of PRR11-amplified MCF7 xenografts is blocked by PI3K inhibition (PMID: 22049316).

3) All of the studies were done with stable cell lines of endocrine resistance or LTED. Do PRR11 levels increase upon acute endocrine treatment or in parental cells? And What are the absolute protein levels of PRR11 across a panel of cell lines representing multiple subtypes of breast cancer?

PRR11 expression was not affected by estrogen deprivation or by tamoxifen treatment for 24 h (data not shown). Moreover, we had shown that E2 stimulation does not induce PRR11 mRNA expression (**Supplementary Fig. 1e** of the revised manuscript).

To address the reviewer's suggestion, we examined PRR11 protein levels in 11 breast cancer cell lines (**Rebuttal Fig. 11A**; included as **Supplementary Fig. 2d** of the revised manuscript). As we replied to Reviewer #1, PRR11 protein levels were higher in PRR11-amplified cell lines. We did not observe a breast cancer subtype-specific expression of PRR11 protein among these cell lines (**Rebuttal Fig. 11A, right**). Further, we interrogated 56 breast cancer cell lines representing all subtypes in the CCLE dataset (PMID: 31068700), and found that PRR11 mRNA expression was similar across subtypes (**Rebuttal Fig. 11B**).

Rebuttal Fig. 11. A. Lysates from breast cancer cell lines (n=11) were subjected to immunoblot analysis with PRR11 and actin antibodies. *PRR11*-amplified cell lines are highlighted in red. The intensity of PRR11 immunoblot bands was quantified using the Image Lab software (ver. 6.0, BioRad) and then plotted by molecular subtype in the right. B. *PRR11* mRNA levels in breast cancer cell lines (n=56). The normalized expression data, log₂ transcript per kilobase million (TPM), was obtained from the CCLE dataset.

4) Figure 4: The results presented show that knockdown of PRR11 results in a S phase entry defect of the cell cycle, demonstrating that PRR11 is needed for S phase entry. In addition, the authors show that there is decrease in CyclinD1 levels. Previous studies have shown that loss of PRR11 results in S phase arrest. This raises question whether PRR11 loss mediated changes in cell cycle is general to all cells and not specific to endocrine resistant variants? Does PRR11 in endocrine sensitive or TNBC cells also result in cell cycle arrest?

As we replied to Reviewer #2, *PRR11* knockdown by siRNA did not suppress proliferation in triple negative and HER2+ breast cancer cells, although it suppressed the AKT phosphorylation (**Rebuttal Fig. 7**). These results imply that *PRR11* does not regulate cell viability through the PIK3/AKT pathway in all cancer cells. Given that *PRR11* has been associated with poor outcome in other various cancer types such as pancreatic, gastric and lung cancers, however, the role of *PRR11* in cell cycle is unlikely to be limited to ER+ breast cancers. Nevertheless, we would like to emphasize that this is the first study demonstrating the role of *PRR11* in anti-estrogen resistance.

5) Figure 5a: it is not clear why knockdown of PRR11 results in a decrease in p110a levels. Is the low pAKT levels simply due to decrease in p110a levels and not related to PRR11?

The interaction of the p85 subunit with the p110 catalytic subunit of PI3K stabilizes p110 α . Since *PRR11* knockdown reduces the p110-p85 interaction as a consequence of interrupting p85 homodimer formation, the p110 α protein would be destabilized upon *PRR11* knockdown. Although we cannot rule out other mechanisms, the lower levels of the p110 α catalytic subunit would result in reduced PI3K activation and p-AKT levels.

6) Figure 5b: It is not clear why overexpression of PRR11 fails increase the levels of p-AKT and p-GSK in MDA-MD-134VI cells. It is certainly not due to difference levels of PRR11 expression or due to endogenous AKT or GSK protein levels. Is this an indication that PRR1 has PI3K independent functions that needs to be addressed?

We respectively disagree that ectopic expression of *PRR11* failed to increase the levels of p-AKT and p-GSK3 β . To clarify this point, however, we reassessed and replaced the data with improved immunoblot images in **Fig. 5B** of the revised manuscript (**Rebuttal Fig. 12**).

Rebuttal Fig. 12. Lysates from MDA-MB-175VII and MDA-MB-134VI cells stably transduced with pLX302-LacZ and -PRR11 were subjected to immunoblot analysis with indicated antibodies.

7) A major conclusion by the authors is that PRR11 interacts with p85 to prevent homodimerization and thus facilitate p85/p110 interaction. The data supporting this model is weak. The stoichiometry of interaction between p85 and PRR11 is not addressed. This is critically because it relates the importance of overexpression and how much overexpression is needed to activate PI3K signaling. It is also not clear if PRR11, p85 and P110 exist in a trimeric complex? What happens to this complex in cells expressing low levels of PRR11 – for eg., parental MCF7 or HCC1428 cells? A careful analysis of stoichiometry of interaction and the impact of overexpression needs to be addressed.

We thank the review for this comment and agree that this experiment will improve our manuscript. To explore the extent of PRR11 expression that is required to activate PI3K, we assessed the impact of gradually increased PRR11 on p85 α homodimerization. We co-transfected pLenti7.3-PIK3R1-Flag, pLenti7.3-PIK3R1-HA and pLX302-PRR11 (0, 0.1, 0.5, 1 μ g) into HEK293 cells. Increased levels of exogenous PRR11 competitively inhibited p85 α homodimerization in a PRR11 dose-dependent fashion (**Rebuttal Fig. 13A**; included as **Fig. 5f** of the revised manuscript). Of note, p85 α homodimers were completely undetectable in cells transfected with the highest amount of pLX302-PRR11. Conversely, we downregulated PRR11 expression (with 0, 1, 5, 25 pM siRNA) in MCF7 LTED cells that had been stably co-transduced with pIND-PIK3R1-HA and pLenti6.3-PIK3R1-Flag. p85 α homodimers increased upon PRR11 knockdown, further suggesting that PRR11 negatively regulates p85 α homodimerization (**Rebuttal Fig. 13B**; included as **Fig. 5g** of the revised manuscript).

Rebuttal Fig. 13. A. HEK293 cells were co-transfected with pLenti7.3-*PIK3R1*-HA, -*PIK3R1*-Flag and pLX302-*PRR11* (0, 0.25, 0.5, 1 μ g) for 48 h. Lysates from these cells were immunoprecipitated with HA antibody. Immune complexes were then subjected to immunoblot analysis with Flag, HA and V5 antibodies. B. MCF7 LTED cells stably co-transduced with pLenti6.3-*PIK3R1*-HA and pIND-*PIK3R1*-Flag were transfected with *PRR11* siRNA (0, 1, 5, 25 pM) for 48 h in presence of 2 μ g/mL doxycycline. Lysates from these cells were immunoprecipitated with HA antibody. Immune complexes were then subjected to immunoblot analysis with Flag, HA and PRR11 antibodies.

8) To rule out PRR11 functions outside of p85 interaction, the authors need to identify PRR11 mutants that will not interact with p85 and assess the impact of overexpressing such a mutant on proliferation and endocrine treatment responsiveness and other assays.

We agree with the reviewer. To address this point, we rescued with either PRR11 wild type (WT) or a with a PRR11 mutant lacking the PR motif (PRR11- Δ PR) in MCF7 LTED cells stably expressing shRNA targeting the 3' UTR of *PRR11*. The PRR11- Δ PR did not induce p-AKT to the same degree as PRR11 WT (**Rebuttal Fig. 14A**; included as **Supplementary Fig. 7c** of the revised manuscript). Consistent with the lower potency of PRR11- Δ PR on AKT activation, rescue with PRR11- Δ PR could not restore cell viability as much as PRR11 WT (**Rebuttal Fig. 14B**).

Rebuttal Fig. 14. A. MCF7 LTED cells were transduced with shRNA targeting the 3' UTR of *PRR11* and then, re-transduced with pLX304-*PRR11* wild type or pLX304-*PRR11* Δ PR. Cell lysates were subjected to immunoblot analysis with V5, p-AKT, AKT and actin antibodies. B. Low density monolayers of cells shown in A were grown in absence of E2 for 10 days. Then, cells were stained with crystal violet and quantified as described in Method (t-test).

9) In Main Fig3 and Supplement Fig3, there is only one sgRNA used for KO of PRR11. Use of an additional sgRNA would make a stronger case that the effects of KO are specific to PRR11.

We had edited MCF7 and HCC1428 cells using 3 independent sgRNAs targeting *PRR11*, which were purchased from Dharmacon (VSGH11937-247492131). Among clones successfully transduced, we selected one clone showing the greatest efficacy in suppression of PRR11 expression and used for the

results shown in **Fig. 3f** and **Extended Data Fig. 3c** of the original manuscript. Since we utilized siRNA, shRNA and sgRNA to knock down *PRR11*, we did not use multiple clones edited by *PRR11* sgRNA. As we replied to Reviewer #1, however, we removed all experiments with *PRR11* sgRNA due to its incomplete knockout efficiency except for experiments with MCF7 29C-1 cells in which *PRR11* sgRNA strongly suppressed *PRR11* expression (**Supplementary Fig. 5b** of the revised manuscript).

10) How does the overexpression of FLAG- and HA-tagged PRR11 compare to endogenous levels of the cell lines used in the manuscript (especially MCF7 LTED)?

To clarify, we did not overexpress *PRR11* tagged with either Flag or HA in this study. Ectopically overexpressed *PRR11* was tagged with only the V5 epitope. In **Fig. 3d** of the revised manuscript, the levels of ectopically expressed *PRR11* were slightly higher than endogenous *PRR11* in both MCF7 and HCC1428 LTED cells.

11) IC50s should be displayed for Fig7e as in Fig3.

To address the concern raised by the Reviewer #2, we have performed growth rate inhibition (GR) metric assay instead of IC_{50} calculation for drug response (**Rebuttal Fig. 8**). Drug response curves shown in **Fig. 3f-h** and **7e** were replaced to plots displaying GR_{50} values in the revised manuscript.

12) Is homodimerization of p85 β similarly blocked in the presence of PRR11? This could play a role in the eventual resistance of the p85 α mechanism.

We thank the reviewer for this insightful point. To our knowledge, however, there is a lack of evidence demonstrating the permissive role of p85 β homodimer in PI3K activation. Considering that p85 β does not act as an inhibitor of PI3K activation (PMID: 22733740) or is a less effective inhibitor of the PI3K catalytic subunit than p85 α (PMID: 25385636), the impact of *PRR11*-mediated p85 β homodimerization on PI3K signaling activation would be expected to be weak or negligible.

Reviewers' Comments:

Reviewer #1:

Remarks to the Author:

The authors have satisfactorily addressed all the concerns on the previous version of the manuscript.

Reviewer #3:

Remarks to the Author:

In this revised manuscript, the authors provide strong data to suggest PRR11 located in 17q23, is critical for conferring endocrine resistance through amplification of PI3K signaling. The results presented suggest that PRR11 interacts with the P85 subunit of PI3K and competitively interferes with homodimerization. Additionally, they demonstrate that PRR11 was both necessary and sufficient to promote proliferation in endocrine resistant/estrogen suppressed ER+ breast cancer cells. Overall, the authors provide strong evidence that PRR11 is a critical mediator of endocrine resistance through regulation of the PI3K pathway. The study is suitable for publication.